# Cross-validation Confidence Intervals for Test Error

**Pierre Bayle**[*]
Princeton University
pbayle@princeton.edu

**Alexandre Bayle**[*]
Harvard University
alexandre_bayle@g.harvard.edu

**Lucas Janson**
Harvard University
ljanson@fas.harvard.edu

**Lester Mackey**
Microsoft Research New England
lmackey@microsoft.com

## Abstract

This work develops central limit theorems for cross-validation and consistent estimators of its asymptotic variance under weak stability conditions on the learning algorithm. Together, these results provide practical, asymptotically-exact confidence intervals for $k$-fold test error and valid, powerful hypothesis tests of whether one learning algorithm has smaller $k$-fold test error than another. These results are also the first of their kind for the popular choice of leave-one-out cross-validation. In our real-data experiments with diverse learning algorithms, the resulting intervals and tests outperform the most popular alternative methods from the literature.

## 1 Introduction

Cross-validation (CV) [49, 26] is a de facto standard for estimating the test error of a prediction rule. By partitioning a dataset into $k$ equal-sized validation sets, fitting a prediction rule with each validation set held out, evaluating each prediction rule on its corresponding held-out set, and averaging the $k$ error estimates, CV produces an unbiased estimate of the test error with lower variance than a single train-validation split could provide. However, these properties alone are insufficient for high-stakes applications in which the uncertainty of an error estimate impacts decision-making. In predictive cancer prognosis and mortality prediction for instance, scientists and clinicians rely on *test error confidence intervals (CIs)* based on CV and other repeated sample splitting estimators to avoid spurious findings and improve reproducibility [42, 45]. Unfortunately, the CIs most often used have no correctness guarantees and can be severely misleading [30]. The difficulty comes from the dependence across the $k$ averaged error estimates: if the estimates were independent, one could derive an *asymptotically-exact CI* (i.e., a CI with coverage converging exactly to the target level) for test error using a standard central limit theorem. However, the error estimates are seldom independent, due to the overlap amongst training sets and between different training and validation sets. Thus, new tools are needed to develop valid, informative CIs based on CV.

The same uncertainty considerations are relevant when comparing two machine learning methods: before selecting a prediction rule for deployment, one would like to be confident that its test error is better than a baseline or an available alternative. The standard practice amongst both method developers and consumers is to conduct a formal *hypothesis test* for a difference in test error between two prediction rules [22, 38, 43, 13, 19]. Unfortunately, the most popular tests from the literature like the cross-validated $t$-test [22], the repeated train-validation $t$-test [43], and the $5 \times 2$ CV test [22] have no correctness guarantees and hence can produce misleading conclusions. The difficulty parallels that of the confidence interval setting: standard tests assume independence and do not

---

[*]Equal contribution

appropriately account for the dependencies across CV error estimates. Therefore, new tools are also needed to develop valid, powerful tests for test error improvement based on CV.

**Our contributions** To meet these needs, we characterize the asymptotic distribution of CV error and develop consistent estimates of its variance under weak stability conditions on the learning algorithm. Together, these results provide practical, asymptotically-exact confidence intervals for test error as well as valid and powerful hypothesis tests of whether one learning algorithm has smaller test error than another. In more detail, we prove in Sec. 2 that $k$-fold CV error is asymptotically normal around its test error under an abstract asymptotic linearity condition. We then give in Sec. 3 two different stability conditions that hold for large classes of learning algorithms and losses and that individually imply the asymptotic linearity condition. In Sec. 4, we propose two estimators of the asymptotic variance of CV and prove them to be consistent under similar stability conditions; our second estimator accommodates any choice of $k$ and appears to be the first consistent variance estimator for leave-one-out CV. To validate our theory in Sec. 5, we apply our intervals and tests to a diverse collection of classification and regression methods on particle physics and flight delay data and observe consistent improvements in width and power over the most popular alternative methods from the literature.

**Related work** Despite the ubiquity of CV, we are only aware of three prior efforts to characterize the precise distribution of cross-validation error. The CV central limit theorem (CLT) of Dudoit and van der Laan [23] requires considerably stronger assumptions than our own and is not paired with the consistent estimate of variance needed to construct a valid confidence interval or test. LeDell et al. [36] derive both a CLT and a consistent estimate of variance for CV, but these apply only to the area under the ROC curve (AUC) performance measure. Finally, in very recent work, Austern and Zhou [5] derive a CLT and a consistent estimate of variance for CV under more stringent assumptions than our own. We compare our results with each of these works in detail in Sec. 3.3. We note also that another work [37] aims to test the difference in test error between two learning algorithms using cross-validation but only proves the validity of their procedure for a single train-validation split rather than for CV. Many other works have studied the problem of bounding or estimating the variance of the cross-validation error [11, 43, 9, 39, 31, 34, 16, 2, 3], but none have established the consistency of their variance estimators. Among these, Kale et al. [31], Kumar et al. [34], Celisse and Guedj [16] introduce relevant notions of algorithmic stability to which we link our results in Sec. 3.1. Moreover, non-asymptotic CIs can be derived from the CV concentration inequalities of [14, 18, 16, 2, 3], but these CIs are more difficult to deploy as they require (1) stronger stability assumptions, (2) a known upper bound on stability, and (3) either a known upper bound on the loss or a known uniform bound on the covariates and a known sub-Gaussianity constant for the response variable. In addition, the reliance on somewhat loose inequalities typically leads to overly large, relatively uninformative CIs. For example, we implemented the ridge regression CI of [16, Thm. 3] for the regression experiment of Sec. 5.1 (see App. K.3). When the features are standardized, the narrowest concentration-based interval is 91 times wider than our widest CLT interval in Fig. 5. Without standardization, the narrowest concentration-based interval is $5 \times 10^{14}$ times wider.

**Notation** Let $\overset{d}{\to}$, $\overset{p}{\to}$, and $\overset{L^q}{\to}$ for $q > 0$, denote convergence in distribution, in probability, and in $L^q$ norm (i.e., $X_n \overset{L^q}{\to} X \Leftrightarrow \mathbb{E}[|X_n - X|^q] \to 0$), respectively. For each $m, n \in \mathbb{N}$ with $m \leq n$, we define the set $[n] \triangleq \{1, \ldots, n\}$ and the vector $m{:}n \triangleq (m, \ldots, n)$. When considering independent random elements $(X, Y)$, we use $\mathbb{E}_X$ and $\text{Var}_X$ to indicate expectation and variance only over $X$, respectively; that is, $\mathbb{E}_X[f(X, Y)] \triangleq \mathbb{E}[f(X, Y) \mid Y]$ and $\text{Var}_X(f(X, Y)) \triangleq \text{Var}(f(X, Y) \mid Y)$. We will refer to the Euclidean norm of a vector as the $\ell^2$ norm in the context of $\ell^2$ regularization.

## 2  A Central Limit Theorem for Cross-validation

In this section, we present a new central limit theorem for $k$-fold cross-validation. Throughout, any asymptotic statement will take $n \to \infty$, and while we allow the number of folds $k_n$ to depend on the sample size $n$ (e.g., $k_n = n$ for leave-one-out cross-validation), we will write $k$ in place of $k_n$ to simplify our notation. We will also present our main results assuming that $k$ evenly divides $n$, but we address the indivisible setting in the appendix.

Hereafter, we will refer to a sequence $(Z_i)_{i \geq 1}$ of random datapoints taking values in a set $\mathcal{Z}$. Notably, $(Z_i)_{i \geq 1}$ need not be independent or identically distributed. We let $Z_{1:n}$ designate the first $n$ points, and, for any vector $B$ of indices in $[n]$, we let $Z_B$ denote the subvector of $Z_{1:n}$ corresponding to ordered indices in $B$. We will also refer to *train-validation splits* $(B, B')$. These are vectors of indices in $[n]$ representing the ordered points assigned to the training set and validation set.[2] As is typical in CV, we will assume that $B$ and $B'$ partition $[n]$, so that every datapoint is either in the training or validation set.

Given a scalar loss function $h_n(Z_i, Z_B)$ and a set of $k$ train-validation splits $\{(B_j, B'_j)\}_{j=1}^k$ with validation indices $\{B'_j\}_{j=1}^k$ partitioning $[n]$ into $k$ folds, we will use the *$k$-fold cross-validation error*

$$\hat{R}_n \triangleq \tfrac{1}{n} \sum_{j=1}^k \sum_{i \in B'_j} h_n(Z_i, Z_{B_j})$$

to draw inferences about the *$k$-fold test error*

$$R_n \triangleq \tfrac{1}{n} \sum_{j=1}^k \sum_{i \in B'_j} \mathbb{E}[h_n(Z_i, Z_{B_j}) \mid Z_{B_j}]. \tag{2.1}$$

A prototypical example of $h_n$ is squared error or 0-1 loss,

$$h_n(Z_i, Z_B) = (Y_i - \hat{f}(X_i; Z_B))^2 \quad \text{or} \quad h_n(Z_i, Z_B) = \mathbb{1}[Y_i \neq \hat{f}(X_i; Z_B)],$$

composed with an algorithm for fitting a prediction rule $\hat{f}(\cdot; Z_B)$ to training data $Z_B$ and predicting the response value of a test point $Z_i = (X_i, Y_i)$.[3] In this setting, the $k$-fold test error is a standard inferential target [11, 23, 31, 34, 5] and represents the average test error of the $k$ prediction rules $\hat{f}(\cdot; Z_{B_j})$. When comparing the performance of two algorithms in Secs. 4 and 5, we will choose $h_n$ to be the difference between the losses of two prediction rules.

## 2.1 Asymptotic linearity of cross-validation

The key to our central limit theorem is establishing that the $k$-fold CV error asymptotically behaves like the $k$-fold test error plus an average of functions applied to single datapoints. The following proposition provides a convenient characterization of this *asymptotic linearity* property.

**Proposition 1** (Asymptotic linearity of $k$-fold CV). *For any sequence of datapoints $(Z_i)_{i \geq 1}$,*

$$\tfrac{\sqrt{n}}{\sigma_n}(\hat{R}_n - R_n) - \tfrac{1}{\sigma_n \sqrt{n}} \sum_{i=1}^n \big(\bar{h}_n(Z_i) - \mathbb{E}\big[\bar{h}_n(Z_i)\big]\big) \xrightarrow{p} \Big(\text{resp.} \xrightarrow{L^q}\Big) 0$$

*for a function $\bar{h}_n$ with $\sigma_n^2 \triangleq \tfrac{1}{n}\mathrm{Var}(\sum_{i=1}^n \bar{h}_n(Z_i))$ if and only if*

$$\tfrac{1}{\sigma_n \sqrt{n}} \sum_{j=1}^k \sum_{i \in B'_j} \Big( h_n\big(Z_i, Z_{B_j}\big) - \mathbb{E}\big[h_n\big(Z_i, Z_{B_j}\big) \mid Z_{B_j}\big] \tag{2.2}$$

$$- \big(\bar{h}_n(Z_i) - \mathbb{E}\big[\bar{h}_n(Z_i)\big]\big) \Big) \xrightarrow{p} \Big(\text{resp.} \xrightarrow{L^q}\Big) 0,$$

*where the parenthetical convergence indicates that the same statement holds when both convergences in probability are replaced with convergences in $L^q$ for the same $q > 0$.*

Typically, one will choose $\bar{h}_n(z) = \mathbb{E}[h_n(z, Z_{1:n(1-1/k)})]$. With this choice, we see that the difference of differences in (2.2) is small whenever $h_n(Z_i, Z_{B_j})$ is close to *either* its expectation given $Z_i$ *or* its expectation given $Z_{B_j}$, but it need not be close to both. As the asymptotic linearity condition (2.2) is still quite abstract, we devote all of Sec. 3 to establishing sufficient conditions for (2.2) that are interpretable, broadly applicable, and simple to verify. Prop. 1 follows from a more general asymptotic linearity characterization for repeated sample-splitting estimators proved in App. A.

## 2.2 From asymptotic linearity to asymptotic normality

So far, we have assumed nothing about the dependencies amongst the datapoints $Z_i$. If we additionally assume that the datapoints are i.i.d., the average $\tfrac{1}{\sigma_n \sqrt{n}} \sum_{i=1}^n \big(\bar{h}_n(Z_i) - \mathbb{E}\big[\bar{h}_n(Z_i)\big]\big)$ converges to a standard normal under a mild integrability condition, and we obtain the following CLT for CV.

**Theorem 1** (Asymptotic normality of $k$-fold CV with i.i.d. data). *In the notation of Prop. 1, suppose that the datapoints $(Z_i)_{i \geq 1}$ are i.i.d. copies of a random element $Z_0$ and that the sequence of $(\bar{h}_n(Z_0) - \mathbb{E}[\bar{h}_n(Z_0)])^2 / \sigma_n^2$ with $\sigma_n^2 = \mathrm{Var}(\bar{h}_n(Z_0))$ is uniformly integrable (UI). If the asymptotic linearity condition (2.2) holds in probability then*

$$\frac{\sqrt{n}}{\sigma_n}(\hat{R}_n - R_n) \xrightarrow{d} \mathcal{N}(0, 1).$$

Thm. 1 is a special case of a more general result, proved in App. B, that applies when the datapoints are independent but not necessarily identically distributed. A simple sufficient condition for the required uniform integrability is that $\sup_n \mathbb{E}[|(\bar{h}_n(Z_0) - \mathbb{E}[\bar{h}_n(Z_0)])/\sigma_n|^\alpha] < \infty$ for some $\alpha > 2$. This holds, for example, whenever $\bar{h}_n(Z_0)$ has uniformly bounded $\alpha$ moments (e.g., the 0-1 loss has all moments uniformly bounded) and does not converge to a degenerate distribution. We now turn our attention to the asymptotic linearity condition.

# 3 Sufficient Conditions for Asymptotic Linearity

## 3.1 Asymptotic linearity from loss stability

Our first result relates the asymptotic linearity of CV to a specific notion of algorithmic stability, termed *loss stability*.

**Definition 1** (Mean-square stability and loss stability). *For $m > 0$, let $Z_0$ and $Z_0', Z_1, \ldots, Z_m$ be i.i.d. test and training points with $Z_{1:m}^{\backslash i}$ representing $Z_{1:m}$ with $Z_i$ replaced by $Z_0'$. For any function $h : \mathcal{Z} \times \mathcal{Z}^m \to \mathbb{R}$, the* mean-square stability *[31] is defined as*

$$\gamma_{ms}(h) \triangleq \frac{1}{m} \sum_{i=1}^m \mathbb{E}[(h(Z_0, Z_{1:m}) - h(Z_0, Z_{1:m}^{\backslash i}))^2] \tag{3.1}$$

*and the* loss stability *[34] as $\gamma_{loss}(h) \triangleq \gamma_{ms}(h')$, where*

$$h'(Z_0, Z_{1:m}) \triangleq h(Z_0, Z_{1:m}) - \mathbb{E}[h(Z_0, Z_{1:m}) \mid Z_{1:m}].$$

Kumar et al. [34] introduced loss stability to bound the variance of CV in terms of the variance of a single hold-out set estimate. Here we show that a suitable decay in loss stability is also sufficient for $L^2$ asymptotic linearity, via a non-asymptotic bound on the departure from linearity.

**Theorem 2** (Approximate linearity from loss stability). *In the notation of Prop. 1 and Def. 1, suppose that the datapoints $(Z_i)_{i \geq 1}$ are i.i.d. copies of a random element $Z_0$. Then*

$$\mathrm{Var}\left(\frac{1}{\sqrt{n}} \sum_{j=1}^k \sum_{i \in B_j'} (h_n'(Z_i, Z_{B_j}) - \mathbb{E}[h_n'(Z_i, Z_{B_j}) \mid Z_i])\right) \leq \frac{3}{2} n \left(1 - \frac{1}{k}\right) \gamma_{loss}(h_n). \tag{3.2}$$

*Hence the $L^2$ asymptotic linearity condition (2.2) holds with $\bar{h}_n(z) = \mathbb{E}[h_n(z, Z_{1:n(1-1/k)})]$ if the loss stability satisfies $\gamma_{loss}(h_n) = o(\sigma_n^2/n)$.*

The proof of Thm. 2 is given in App. C. Recall that in a typical learning context, we have $h_n(Z_0, Z_{1:m}) = \ell(Y_0, \hat{f}(X_0; Z_{1:m}))$ for a fixed loss $\ell$, a learned prediction rule $\hat{f}(\cdot; Z_{1:m})$, a test point $Z_0 = (X_0, Y_0)$, and $m = n(1 - 1/k)$. When $\hat{f}(\cdot; Z_{1:m})$ converges to an imperfect prediction rule, we will commonly have $\sigma_n^2 = \mathrm{Var}(\mathbb{E}[h_n(Z_0, Z_{1:m}) \mid Z_0]) = \Omega(1)$ so that $\gamma_{loss}(h_n) = o(1/n)$ loss stability is sufficient. However, Thm. 2 also accommodates the cases of non-convergent $\hat{f}(\cdot; Z_{1:m})$ and of $\hat{f}(\cdot; Z_{1:m})$ converging to a perfect prediction rule, so that $\sigma_n^2 = o(1)$.

Many learning algorithms are known to enjoy decaying loss stability [14, 25, 29, 16, 4], in part because loss stability is upper-bounded by a variety of algorithmic stability notions studied in the literature. For example, stochastic gradient descent on convex and non-convex objectives [29] and the empirical risk minimization of a strongly convex and Lipschitz objective both have $O(1/n)$ *uniform stability* [14] which implies a loss stability of $O(1/n^2) = o(1/n)$ by [31, Lem. 1] and [34, Lem. 2]. However, we emphasize that the loss $h_n$ need not be convex and need not coincide with a loss function used to train a learning method. Indeed, our stability assumptions also cover $k$-nearest neighbor methods [20], decision tree methods [4], and ensemble methods [25] and can even hold when training error is a poor proxy for test error due to overfitting (e.g., 1-nearest neighbor has training error 0 but is still suitably stable [20]). In addition, for any loss function, loss stability is upper-bounded by mean-square stability [31] and all $L^q$ stabilities [16] for $q \geq 2$. For bounded loss functions such as the 0-1 loss, loss stability is also weaker than hypothesis stability (also called $L^1$ stability) [20, 32], weak-hypothesis stability [21], and weak-$L^1$ stability [35].

## 3.2 Asymptotic linearity from conditional variance convergence

We can also guarantee asymptotic linearity under weaker moment conditions than Thm. 2 at the expense of stronger requirements on the number of folds $k$.

**Theorem 3** (Asymptotic linearity from conditional variance convergence). *In the notation of Prop. 1, suppose that the datapoints $(Z_i)_{i \geq 1}$ are i.i.d. copies of a random element $Z_0$. If*

$$\max(k^{q/2}, k^{1-q/2})\mathbb{E}\left[\left(\tfrac{1}{\sigma_n^2}\mathrm{Var}_{Z_0}\left(h_n(Z_0, Z_{1:n(1-1/k)}) - \bar{h}_n(Z_0))\right)^{q/2}\right] \to 0 \qquad (3.3)$$

*for a function $\bar{h}_n$ and $q \in (0, 2]$, then $\bar{h}_n$ satisfies the $L^q$ asymptotic linearity condition (2.2). If*

$$\mathbb{E}\left[\min\left(k, \tfrac{\sqrt{k}}{\sigma_n}\sqrt{\mathrm{Var}_{Z_0}\left(h_n(Z_0, Z_{1:n(1-1/k)}) - \bar{h}_n(Z_0)\right)}\right)\right] \to 0. \qquad (3.4)$$

*for a function $\bar{h}_n$, then $\bar{h}_n$ satisfies the in-probability asymptotic linearity condition (2.2).*

**Remark 1.** *When $k = O(1)$, (3.4) holds $\Leftrightarrow \tfrac{1}{\sigma_n}\sqrt{\mathrm{Var}_{Z_0}\left(h_n(Z_0, Z_{1:n(1-1/k)}) - \bar{h}_n(Z_0)\right)} \xrightarrow{p} 0$.*

Thm. 3 follows from a more general statement proved in App. D. When $k$ is bounded, as in 10-fold CV, the conditions of Thm. 3 are considerably weaker than those of Thm. 2 (see App. E), granting asymptotic linearity whenever the conditional variance converges in probability rather than in $L^2$. Indeed in App. G, we detail a simple learning problem in which the loss stability is infinite but Thms. 1 and 3 together provide a valid CLT with convergent variance $\sigma_n^2$.

## 3.3 Comparison with prior work

Our sufficient conditions for asymptotic normality are significantly less restrictive and more broadly applicable than the three prior distributional characterizations of CV error [23, 36, 5]. In particular, the CLT of Dudoit and van der Laan [23, Thm. 3] assumes a bounded loss function, excludes the popular case of leave-one-out cross-validation, and requires the prediction rule to be loss-consistent for a risk-minimizing prediction rule. Similarly, the CLT of LeDell et al. [36, Thm. 4.1] applies only to AUC loss, requires the prediction rule to be loss-consistent for a deterministic prediction rule, and requires a bounded number of folds.

Moreover, in our notation, the recent CLT of Austern and Zhou [5, Thm. 1] restricts focus to learning algorithms that treat all training points symmetrically, assumes that its variance parameter

$$\tilde{\sigma}_n^2 \triangleq \mathbb{E}[\mathrm{Var}(h_n(Z_0, Z_{1:m}) \mid Z_{1:m})] \qquad (3.5)$$

converges to a non-zero limit, requires mean-square stability $\gamma_{ms}(h_n) = o(1/n)$, and places a $o(1/n^2)$ constraint on the second-order mean-square stability

$$\mathbb{E}[((h_n(Z_0, Z_{1:m}) - h_n(Z_0, Z_{1:m}^{\backslash 1})) - (h_n(Z_0, Z_{1:m}^{\backslash 2}) - h_n(Z_0, Z_{1:m}^{\backslash 1,2})))^2] = o(1/n^2), \qquad (3.6)$$

where $Z_{1:m}^{\backslash 1,2}$ represents $Z_{1:m}$ with $Z_1, Z_2$ replaced by i.i.d. copies $Z_1', Z_2'$. Kumar et al. [34] showed that the mean-square stability is always an upper bound for the loss stability required by our Thm. 2, and in Apps. F and G we exhibit two simple learning tasks in which $\gamma_{loss}(h_n) = O(1/n^2)$ but $\gamma_{ms}(h_n) = \infty$. Furthermore, when $k$ is constant, as in 10-fold CV, our conditional variance assumptions in Sec. 3.2 are weaker still and hold even for algorithms with infinite loss stability (see App. G). In addition, our results allow for asymmetric learning algorithms (like stochastic gradient descent), accommodate growing, vanishing, and non-convergent variance parameters $\sigma_n^2$, and do not require the second-order mean-square stability condition (3.6).

Finally, we note that the asymptotic variance parameter $\sigma_n^2$ appearing in Thm. 1 is never larger and sometimes smaller than the variance parameter $\tilde{\sigma}_n^2$ in [5, Thm. 1].

**Proposition 2** (Variance comparison). *Let $\sigma_n^2 = \mathrm{Var}(\mathbb{E}[h_n(Z_0, Z_{1:m}) \mid Z_0])$ be the variance appearing in Thm. 1, with the choice $\bar{h}_n(z) = \mathbb{E}[h_n(z, Z_{1:m})]$, and $\tilde{\sigma}_n^2 = \mathbb{E}[\mathrm{Var}(h_n(Z_0, Z_{1:m}) \mid Z_{1:m})]$ be the variance parameter of [5, Eq. (15)] for $m = n(1 - 1/k)$. Then*

$$\sigma_n^2 \leq \tilde{\sigma}_n^2 \leq \sigma_n^2 + \tfrac{m}{2}\gamma_{loss}(h_n),$$

*and the first inequality is strict whenever $h(Z_0, Z_{1:m}) - \mathbb{E}[h(Z_0, Z_{1:m}) \mid Z_{1:m}]$ depends on $Z_{1:m}$.*

The proof of Prop. 2 can be found in App. H. In App. G, we present a simple learning task for which our central limit theorem provably holds with $\sigma_n^2$ converging to a non-zero constant, but the CLT in [5, Eq. (15)] is inapplicable because the variance parameter $\tilde{\sigma}_n^2$ is infinite.

# 4 Confidence Intervals and Tests for $k$-fold Test Error

A primary application of our central limit theorems is the construction of asymptotically-exact confidence intervals for the unknown $k$-fold test error. For example, under the assumptions and notation of Thm. 1, any sample statistic $\hat{\sigma}_n^2$ satisfying relative error consistency, $\hat{\sigma}_n^2/\sigma_n^2 \xrightarrow{p} 1$, gives rise to an asymptotically-exact $(1-\alpha)$-confidence interval,

$$C_\alpha \triangleq \hat{R}_n \pm q_{1-\alpha/2}\hat{\sigma}_n/\sqrt{n} \quad \text{satisfying} \quad \lim_{n\to\infty} \mathbb{P}(R_n \in C_\alpha) = 1 - \alpha, \qquad (4.1)$$

where $q_{1-\alpha/2}$ is the $(1-\alpha/2)$-quantile of a standard normal distribution.

A second, related application of our central limit theorems is testing whether, given a dataset $Z_{1:n}$, a $k$-fold partition $\{B_j'\}_{j=1}^k$, and two algorithms $\mathcal{A}_1$, $\mathcal{A}_2$ for fitting prediction rules, $\mathcal{A}_2$ has larger $k$-fold test error than $\mathcal{A}_1$. In this circumstance, we may define

$$h_n(Z_0, Z_B) = \ell(Y_0, \hat{f}_1(X_0; Z_B)) - \ell(Y_0, \hat{f}_2(X_0; Z_B))$$

to be the difference of the loss functions of two prediction rules trained on $Z_B$ and tested on $Z_0 = (X_0, Y_0)$. Our aim is to test whether $\mathcal{A}_1$ improves upon $\mathcal{A}_2$ on the fold partition, that is to test the null $H_0 : R_n \geq 0$ against the alternative hypothesis $H_1 : R_n < 0$. Under the assumptions and notation of Thm. 1, an asymptotically-exact level-$\alpha$ test is given by[4]

$$\text{REJECT } H_0 \Leftrightarrow \hat{R}_n < q_\alpha \hat{\sigma}_n/\sqrt{n} \qquad (4.2)$$

where $q_\alpha$ is the $\alpha$-quantile of a standard normal distribution and $\hat{\sigma}_n^2$ is any variance estimator satisfying relative error consistency, $\hat{\sigma}_n^2/\sigma_n^2 \xrightarrow{p} 1$. Fortunately, our next theorem describes how to compute such a consistent estimate of $\sigma_n^2$ under weak conditions.

**Theorem 4** (Consistent within-fold estimate of asymptotic variance). *In the notation of Thm. 1 with $m = n(1-1/k)$, $\bar{h}_n(z) = \mathbb{E}[h_n(z, Z_{1:m})]$, and $k < n$, define the within-fold variance estimator*

$$\hat{\sigma}_{n,in}^2 \triangleq \frac{1}{k}\sum_{j=1}^k \frac{1}{(n/k)-1}\sum_{i\in B_j'}\left(h_n(Z_i, Z_{B_j}) - \frac{k}{n}\sum_{i'\in B_j'}h_n(Z_{i'}, Z_{B_j})\right)^2.$$

*Suppose $(Z_i)_{i\geq 1}$ are i.i.d. copies of a random element $Z_0$. Then $\hat{\sigma}_{n,in}^2/\sigma_n^2 \xrightarrow{L^1} 1$ whenever $\gamma_{loss}(h_n) = o(\sigma_n^2/n)$ and the sequence of $(\bar{h}_n(Z_0) - \mathbb{E}[\bar{h}_n(Z_0)])^2/\sigma_n^2$ is uniformly integrable (UI). Moreover, $\hat{\sigma}_{n,in}^2/\sigma_n^2 \xrightarrow{L^2} 1$ whenever $\mathbb{E}[((\bar{h}_n(Z_0) - \mathbb{E}[\bar{h}_n(Z_0)])/\sigma_n)^4] = o(n)$ and the fourth-moment loss stability $\gamma_4(h_n') \triangleq \frac{1}{m}\sum_{i=1}^m \mathbb{E}[(h_n'(Z_0, Z_{1:m}) - h_n'(Z_0, Z_{1:m}^{\setminus i}))^4] = o(\sigma_n^4/n^2)$. Here, $Z_{1:m}^{\setminus i}$ denotes $Z_{1:m}$ with $Z_i$ replaced by an identically distributed copy independent of $Z_{0:m}$.*

Thm. 4 follows from explicit error bounds proved in App. I. A notable take-away is that the same two conditions—loss stability $\gamma_{loss}(h_n) = o(\sigma_n^2/n)$ and a UI sequence of $(\bar{h}_n(Z_0) - \mathbb{E}[\bar{h}_n(Z_0)])^2/\sigma_n^2$—grant both a central limit theorem for CV (by Thms. 1 and 2) and an $L^1$-consistent estimate of $\sigma_n^2$ (by Thm. 4). Moreover, the $L^2$-consistency bound of App. I can be viewed as a strengthening of the consistency result of [5, Prop. 1] which analyzes the same variance estimator under more stringent assumptions. In our notation, to establish $L^2$ consistency, [5, Prop. 1] additionally requires $h_n$ symmetric in its training points, convergence of the variance parameter $\tilde{\sigma}_n^2$ (3.5) to a non-zero constant, control over a fourth-moment analogue of mean-square stability $\gamma_4(h_n) = o(\sigma_n^4/n^2)$ instead of the smaller fourth-moment loss stability $\gamma_4(h_n')$, and the more restrictive fourth-moment condition $\mathbb{E}[(h_n(Z_0, Z_{1:m})/\sigma_n)^4] = O(1)$.[5] By Prop. 2, their assumptions further imply that $\sigma_n^2$ converges to a non-zero constant. In contrast, Thm. 4 accommodates growing, vanishing, and non-convergent variance parameters $\sigma_n^2$ and a wider variety of learning procedures and losses.

Since Thm. 4 necessarily excludes the case of leave-one-out CV ($k = n$), we propose a second estimator with consistency guarantees for any $k$ and only slightly stronger stability conditions than Thm. 4 when $k = \Omega(n)$. Notably, Austern and Zhou [5] do not provide a consistent variance estimator for $k = n$, and Dudoit and van der Laan [23] do not establish the consistency of any variance estimator.

**Theorem 5** (Consistent all-pairs estimate of asymptotic variance). *Under the notation of Thm. 1 with $m = n(1 - 1/k)$, and $\bar{h}_n(z) = \mathbb{E}[h_n(z, Z_{1:m})]$, define the all-pairs variance estimator*

$$\hat{\sigma}_{n,out}^2 \triangleq \frac{1}{k} \sum_{j=1}^{k} \frac{k}{n} \sum_{i \in B_j'} (h_n(Z_i, Z_{B_j}) - \hat{R}_n)^2.$$

*If $(Z_i)_{i \geq 1}$ are i.i.d. copies of a random element $Z_0$, then $\hat{\sigma}_{n,out}^2 / \sigma_n^2 \xrightarrow{L^1} 1$ whenever $\gamma_{loss}(h_n) = o(\sigma_n^2/n)$, $\gamma_{ms}(h_n) = o(k\sigma_n^2/n)$, and the sequence of $(\bar{h}_n(Z_0) - \mathbb{E}[\bar{h}_n(Z_0)])^2/\sigma_n^2$ is UI.*

Thm. 5 follows from an explicit error bound proved in App. J and differs from the $L^1$-consistency result of Thm. 4 only in the added requirement $\gamma_{ms}(h_n) = o(k\sigma_n^2/n)$. This mean-square stability condition is especially mild when $k = \Omega(n)$ (as in the case of leave-one-out CV) and ensures that two training sets differing in only $n/k$ points produce prediction rules with comparable test losses.

Importantly, both $\hat{\sigma}_{n,in}^2$ and $\hat{\sigma}_{n,out}^2$ can be computed in $O(n)$ time using just the individual datapoint losses $h_n(Z_i, Z_{B_j})$ outputted by a run of $k$-fold cross-validation. Moreover, when $h_n$ is binary, as in the case of 0-1 loss, one can compute $\hat{\sigma}_{n,out}^2 = \hat{R}_n(1 - \hat{R}_n)$ in $O(1)$ time given access to the overall cross-validation error $\hat{R}_n$ and $\hat{\sigma}_{n,in}^2 = \frac{1}{k} \sum_{j=1}^{k} \frac{(n/k)}{(n/k)-1} \hat{R}_{n,j}(1 - \hat{R}_{n,j})$ in $O(k)$ time given access to the $k$ average fold errors $\hat{R}_{n,j} \triangleq \frac{k}{n} \sum_{i \in B_j'} h_n(Z_i, Z_{B_j})$.

## 5 Numerical Experiments

In this section, we compare our test error confidence intervals (4.1) and tests for algorithm improvement (4.2) with the most popular alternatives from the literature: the hold-out test described in [5, Eq. (17)] based on a single train-validation split, the cross-validated $t$-test [22], the repeated train-validation $t$-test [43] (with and without correction), and the $5 \times 2$-fold CV test [22].[6] These procedures are commonly used and admit both two-sided CIs and one-sided tests, but, unlike our proposals, none except the hold-out method are known to be valid. Our aim is to verify whether our proposed procedures outperform these popular heuristics across a diversity of settings encountered in real learning problems. We fix $k = 10$, use 90-10 train-validation splits for all tests save $5 \times 2$-fold CV, and report our results using $\hat{\sigma}_{n,out}^2$ (as $\hat{\sigma}_{n,in}^2$ results are nearly identical).

Evaluating the quality of CIs and tests requires knowledge of the target test error.[7] In each experiment, we use points subsampled from a large real dataset to form a surrogate ground-truth estimate of the test error. Then, we evaluate the CIs and tests constructed from 500 training sets of sample sizes $n$ ranging from 700 to $11,000$ subsampled from the same dataset. Each mean width estimate is displayed with a $\pm 2$ standard error confidence band. The surrounding confidence bands for the coverage, size, and power estimates are 95% Wilson intervals [51], which are known to provide more accurate coverage for binomial proportions than a $\pm 2$ standard error interval [15]. We use the `Higgs` dataset of [6, 7] to study the classification error of random forest, neural network, and $\ell^2$-penalized logistic regression classifiers and the Kaggle `FlightDelays` dataset of [1] to study the mean-squared regression error of random forest, neural network, and ridge regression. In each case, we focus on stable settings of these learning algorithms with sufficiently strong $\ell^2$ regularization for the neural network, logistic, and ridge learners and small depths for the random forest trees. Complete experimental details are available in App. K.1, and code replicating all experiments can be found at https://github.com/alexandre-bayle/cvci.

### 5.1 Confidence intervals for test error

In App. L.1, we compare the coverage and width of each procedure's 95% CI for each of the described algorithms, datasets, and training set sizes. Two representative examples—logistic regression classification and random forest regression—are displayed in Fig. 1. While the repeated train-validation CI significantly undercovers in all cases, all remaining CIs have coverage near the 95%

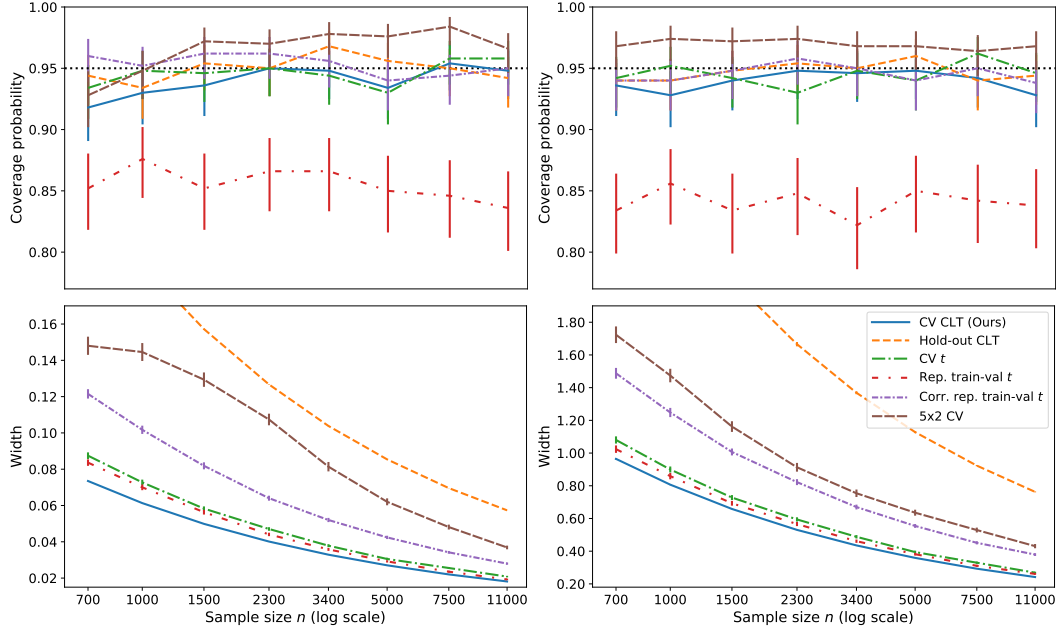

Figure 1: Test error coverage (top) and width (bottom) of $95\%$ confidence intervals (see Sec. 5.1). **Left:** $\ell^2$-regularized logistic regression classifier. **Right:** Random forest regression.

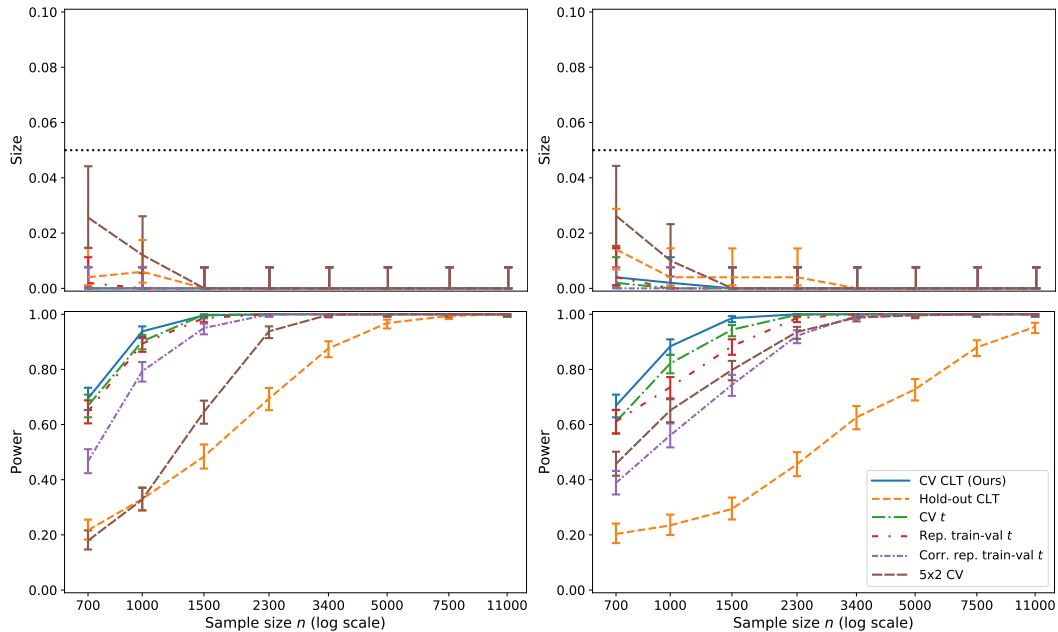

Figure 2: Size when testing $H_1 : \text{Err}(\mathcal{A}_1) < \text{Err}(\mathcal{A}_2)$ (top) and power when testing $H_1 : \text{Err}(\mathcal{A}_2) < \text{Err}(\mathcal{A}_1)$ (bottom) of level-0.05 tests for improved test error (see Sec. 5.2). **Left**: $\mathcal{A}_1 = \ell^2$-regularized logistic regression, $\mathcal{A}_2 = $ neural network classification. **Right**: $\mathcal{A}_1 = $ random forest, $\mathcal{A}_2 = $ ridge regression.

target, even for the smallest training set size of $n = 700$. The hold-out CI, while valid, is substantially wider and less informative than the other intervals as it is based on only a single train-validation split. Meanwhile, our CLT-based CI delivers the smallest width[8] (and hence greatest precision) for both learning tasks and every dataset size.

## 5.2 Testing for improved algorithm performance

Let us write $\text{Err}(\mathcal{A}_1) < \text{Err}(\mathcal{A}_2)$ to signify that the test error of $\mathcal{A}_1$ is smaller than that of $\mathcal{A}_2$. In App. L.2, for each testing procedure, dataset, and pair of algorithms $(\mathcal{A}_1, \mathcal{A}_2)$, we display the size and power of level $\alpha = 0.05$ one-sided tests (4.2) of $H_1 : \text{Err}(\mathcal{A}_1) < \text{Err}(\mathcal{A}_2)$. In each case, we report size estimates for experiments with at least 25 replications under the null and power estimates for experiments with at least 25 replications under the alternative. Here, for representative algorithm pairs, we identify the algorithm $\mathcal{A}_1$ that more often has smaller test error across our simulations and display both the power of the level $\alpha = 0.05$ test of $H_1 : \text{Err}(\mathcal{A}_1) < \text{Err}(\mathcal{A}_2)$ and the size of the level $\alpha = 0.05$ test of $H_1 : \text{Err}(\mathcal{A}_2) < \text{Err}(\mathcal{A}_1)$. Fig. 2 displays these results for $(\mathcal{A}_1, \mathcal{A}_2) = (\ell^2$-regularized logistic regression, neural network) classification on the left and $(\mathcal{A}_1, \mathcal{A}_2) = $ (random forest, ridge) regression on the right. The sizes of all testing procedures are below the nominal level of 0.05, and our test is consistently the most powerful for both classification and regression. The hold-out test, while also valid, is significantly less powerful due to its reliance on a single train-validation split. In App. L.3, we observe analogous results when labels are synthetically generated.

## 5.3 The importance of stability

To illustrate the impact of algorithmic instability on testing procedures, we additionally compare a less stable neural network (with substantially reduced $\ell^2$ regularization strength) and a less stable random forest regressor (with larger-depth trees). In Fig. 13 in App. L.4, we observe that the size of every test save the hold-out test rises above the nominal level. In the case of our test, the cause of this size violation is clear. Fig. 15a in App. L.4 demonstrates that the variance of $\frac{\sqrt{n}}{\sigma_n}(\hat{R}_n - R_n)$ in Thm. 1 is much larger than 1 for this experiment, and Thm. 2 implies this can only occur when the loss stability $\gamma_{loss}(h_n)$ is large. Meanwhile, the variance of the same quantity is close to 1 for the original stable settings of the neural network and random forest regressors. We suspect that instability is also the cause of the other tests' size violations; however, it is difficult to be certain, as these alternative tests have no correctness guarantees. Interestingly, the same destabilized algorithms produce high-quality confidence intervals and relatively stable $h_n$ in the context of single algorithm assessment (see Figs. 14 and 15b in App. L.4), as the variance parameter $\sigma_n^2 = \text{Var}(\bar{h}_n(Z_0))$ is significantly larger for single algorithms. This finding highlights an important feature of our results: it suffices for the loss stability to be negligible relative to the noise level $\sigma_n^2/n$.

## 5.4 Leave-one-out cross-validation

Leave-one-out cross-validation (LOOCV) is often viewed as prohibitive for large datasets, due to the expense of refitting a prediction rule $n$ times. However, for ridge regression, a well-known shortcut based on the Sherman–Morrison–Woodbury formula allows one to carry out LOOCV exactly in the time required to fit a small number of base ridge regressions (see App. K.4 for a derivation of this result). Moreover, recent work shows that, for many learning procedures, LOOCV estimates can be efficiently approximated with only $O(1/n^2)$ error [8, 28, 33, 50] (see also [46, 48, 27] for related guarantees). The $O(1/n^2)$ precision of these inexpensive approximations coupled with the LOOCV consistency of $\hat{\sigma}_{n,out}^2$ (see Thm. 5) allows us to efficiently construct asymptotically-valid CIs and tests for LOOCV, even when $n$ is large. As a simple demonstration, we construct 95% CIs for ridge regression test error based on our LOOCV CLT and compare their coverage and width with those of the procedures described in Sec. 5.1. In Fig. 3 in App. K.4, we see that, like the 10-fold CV CLT intervals, the LOOCV intervals provide coverage near the nominal level and widths smaller than the popular alternatives from the literature; in fact, the 10-fold CV CLT curves are obscured by the nearly identical LOOCV CLT curves. Complete experimental details can be found in App. K.4.

## 6 Conclusion and Future Work

Our central limit theorems and consistent variance estimators provide new, valid tools for testing algorithm improvement and generating test error intervals under algorithmic stability. An important open question is whether practical valid tests and intervals are also available when our stability conditions are violated. Another promising direction for future work is developing analogous tools for the *expected* test error $\mathbb{E}[R_n]$ instead of the $k$-fold test error $R_n$; Austern and Zhou [5] provide significant progress in this direction, but more work, particularly on variance estimation, is needed.

## Broader Impact

This work will benefit both users and developers of machine learning methods who want to rigorously assess or compare learning algorithms. Failure of the methods we discuss (which can only happen when the assumptions we state are not satisfied) may lead to the over- or under-estimation of the performance of a learning algorithm on a particular dataset.

## Acknowledgments and Disclosure of Funding

We would like to thank Jianqing Fan, Mykhaylo Shkolnikov, Miklos Racz, and Morgane Austern for helpful discussions.

## Footnotes

[2]We keep track of index order to support asymmetric learning algorithms like stochastic gradient descent.

[3]For randomized learning algorithms (such as random forests or stochastic gradient descent), all statements in this paper should be treated as holding conditional on the external source of randomness.

[4]The test (4.2) is equivalent to rejecting when the one-sided interval $(-\infty, \hat{R}_n - q_\alpha \hat{\sigma}_n/\sqrt{n}\,]$ excludes 0.

[5]The result [5, Prop. 1] also assumes a fourth moment second-order stability condition similar to (3.6), but this appears to not be used in the proof.

[6]We exclude McNemar's test [40] and the difference-of-proportions test which Dietterich [22] found to be less powerful than $5 \times 2$-fold CV and the conservative $Z$-test which Nadeau and Bengio [43] found less powerful and more expensive than corrected repeated train-validation splitting.

[7]Generalizing the notion of $k$-fold test error (2.1), we define the target test error for each testing procedure to be the average test error of the learned prediction rules; see App. K.2 for more details.

[8]All widths in Fig. 1 are displayed with $\pm 2$ standard error bars, but some bars are too small to be visible.

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
