[Supplementary Material]

# A Proof of Prop. 1: Asymptotic linearity of $k$-fold CV

We first prove a general asymptotic linearity result for repeated sample-splitting estimators. Given a collection $A_n = \{(B_j, B_j')\}_{j \in [J]}$ of index vector pairs such that for any pair $(B_j, B_j')$ in $A_n$, $B_j$ and $B_j'$ are disjoint, and a scalar loss function $\rho_{n,j}(Z_{B_j'}, Z_{B_j})$, define the *cross-validation error* as

$$\hat{R}_n = \tfrac{1}{J} \sum_{j=1}^{J} \rho_{n,j}(Z_{B_j'}, Z_{B_j})$$

and the *multi-fold test error*

$$R_n = \tfrac{1}{J} \sum_{j=1}^{J} \mathbb{E}[\rho_{n,j}(Z_{B_j'}, Z_{B_j}) \mid Z_{B_j}].$$

Note that similarly to the number of folds $k$ in cross-validation, $J$ can depend on the sample size $n$, but we write $J$ in place of $J_n$ to simplify our notation.

**Proposition 3** (Asymptotic linearity of CV). *For any sequence of datapoints $(Z_i)_{i \geq 1}$,*

$$\tfrac{\sqrt{n}}{\sigma_n}\left(\hat{R}_n - R_n\right) - \tfrac{\sqrt{n}}{\sigma_n J} \sum_{j=1}^{J} (\bar{\rho}_{n,j}(Z_{B_j'}) - \mathbb{E}[\bar{\rho}_{n,j}(Z_{B_j'})]) \xrightarrow{p} \left(resp. \xrightarrow{L^q}\right) 0$$

*for functions $\bar{\rho}_{n,1}, \ldots, \bar{\rho}_{n,J}$ with $\sigma_n^2 \triangleq \tfrac{1}{J}\mathrm{Var}(\sum_{j=1}^{J} \bar{\rho}_{n,j}(Z_{B_j'}))$ if and only if*

$$\tfrac{\sqrt{n}}{\sigma_n J} \sum_{j=1}^{J} \Big( \rho_{n,j}(Z_{B_j'}, Z_{B_j}) - \mathbb{E}\Big[\rho_{n,j}(Z_{B_j'}, Z_{B_j}) \mid Z_{B_j}\Big]$$
$$- \Big(\bar{\rho}_{n,j}(Z_{B_j'}) - \mathbb{E}\Big[\bar{\rho}_{n,j}(Z_{B_j'})\Big]\Big)\Big) \xrightarrow{p} \left(resp. \xrightarrow{L^q}\right) 0$$

*where the parenthetical convergence indicates that the same statement holds when both convergences in probability are replaced with convergences in $L^q$ for the same $q > 0$.*

**Proof** For each $(B_j, B_j') \in A_n$, let

$$L_j = \rho_{n,j}(Z_{B_j'}, Z_{B_j}) - \mathbb{E}[\rho_{n,j}(Z_{B_j'}, Z_{B_j}) \mid Z_{B_j}] - \Big(\bar{\rho}_{n,j}(Z_{B_j'}) - \mathbb{E}[\bar{\rho}_{n,j}(Z_{B_j'})]\Big).$$

Then

$$\tfrac{\sqrt{n}}{\sigma_n}\left(\hat{R}_n - R_n\right) = \tfrac{\sqrt{n}}{\sigma_n J} \sum_{j=1}^{J} (\rho_{n,j}(Z_{B_j'}, Z_{B_j}) - \mathbb{E}[\rho_{n,j}(Z_{B_j'}, Z_{B_j}) \mid Z_{B_j}])$$
$$= \tfrac{\sqrt{n}}{\sigma_n J} \sum_{j=1}^{J} L_j + \tfrac{\sqrt{n}}{\sigma_n J} \sum_{j=1}^{J} (\bar{\rho}_{n,j}(Z_{B_j'}) - \mathbb{E}[\bar{\rho}_{n,j}(Z_{B_j'})]).$$

The result now follows from the assumption that $\tfrac{\sqrt{n}}{\sigma_n J} \sum_{j=1}^{J} L_j \xrightarrow{p} \left(\text{resp. } \xrightarrow{L^q}\right) 0$. □

Prop. 1 now follows directly from Prop. 3 with the choices:

- $A_n = \{(B_\ell, i) : \ell \in [k], i \in B_\ell'\}$,
- for all $j \in [J]$, $\rho_{n,j}(Z_i, Z_{B_\ell}) = h_n(Z_i, Z_{B_\ell})$ and $\bar{\rho}_{n,j}(Z_i) = \bar{h}_n(Z_i)$ for the associated $\ell \in [k]$ and $i \in B_\ell'$.

Note that for these choices, we have $J = |A_n| = \sum_{\ell=1}^{k} |B_\ell'| = n$.

# B Proof of Thm. 1: Asymptotic normality of $k$-fold CV with i.i.d. data

Thm. 1 follows from the next more general result, which establishes the asymptotic normality of $k$-fold CV with independent (not necessarily identically distributed) data.

**Theorem 6** (Asymptotic normality of $k$-fold CV with independent data). *Under the notation of Prop. 1, suppose that the datapoints $(Z_i)_{i \geq 1}$ are independent. If the triangular array $\big(\bar{h}_n(Z_i) - \mathbb{E}\big[\bar{h}_n(Z_i)\big]\big)_{n,i}$ satisfies* Lindeberg's condition,

$$\forall \varepsilon > 0, \tfrac{1}{n\sigma_n^2} \sum_{i=1}^{n} \mathbb{E}\Big[\big(\bar{h}_n(Z_i) - \mathbb{E}\big[\bar{h}_n(Z_i)\big]\big)^2 \mathbb{1}\big[|\bar{h}_n(Z_i) - \mathbb{E}\big[\bar{h}_n(Z_i)\big]| > \varepsilon\, \sigma_n \sqrt{n}\big]\Big] \to 0, \text{ (B.1)}$$

*then*

$$\frac{1}{\sigma_n \sqrt{n}} \sum_{i=1}^{n} \left(\bar{h}_n(Z_i) - \mathbb{E}\left[\bar{h}_n(Z_i)\right]\right) \xrightarrow{d} \mathcal{N}(0,1).$$

*Additionally, if (2.2) holds in probability, then*

$$\frac{\sqrt{n}}{\sigma_n}\left(\hat{R}_n - R_n\right) \xrightarrow{d} \mathcal{N}(0,1).$$

**Proof**    By independence of the datapoints $(Z_i)_{i \geq 1}$, $(\bar{h}_n(Z_i))_{n,i}$ are independent, and $n\sigma_n^2 = \mathrm{Var}(\sum_{i=1}^{n} \bar{h}_n(Z_i))$. Under Lindeberg's condition, we get the first convergence result thanks to Lindeberg's Central Limit Theorem (see [10, Thm. 27.2]). Additionally, if assumption (2.2) holds, we apply Prop. 1 and Slutsky's theorem to get the second convergence result. □

If the $(Z_i)_{i \geq 1}$ are i.i.d., then $\sigma_n^2 = \frac{1}{n}\mathrm{Var}(\sum_{i=1}^{n} \bar{h}_n(Z_i)) = \mathrm{Var}(\bar{h}_n(Z_0))$, and Lindeberg's condition (B.1) reduces to

$$\forall \varepsilon > 0, \frac{1}{\sigma_n^2}\mathbb{E}\left[\left(\bar{h}_n(Z_0) - \mathbb{E}\left[\bar{h}_n(Z_0)\right]\right)^2 \mathbb{1}\left[|\bar{h}_n(Z_0) - \mathbb{E}\left[\bar{h}_n(Z_0)\right]| > \varepsilon\,\sigma_n\sqrt{n}\right]\right] \to 0.$$

We will show that this follows from the assumed uniform integrability of the sequence $X_n = (\bar{h}_n(Z_0) - \mathbb{E}[\bar{h}_n(Z_0)])^2/\sigma_n^2$. Indeed, for any $\varepsilon > 0$ and all $n$,

$$\mathbb{E}[X_n \mathbb{1}[X_n > n\varepsilon^2]] \leq \sup_m \mathbb{E}[X_m \mathbb{1}[X_m > n\varepsilon^2]] \to 0,$$

as $n \to \infty$ by the uniform integrability of the sequence of $X_n$. Thm. 1 therefore follows from Thm. 6.

## C    Proof of Thm. 2: Approximate linearity from loss stability

Thm. 2 will follow from the following more general result.

**Theorem 7** (Approximate linearity from loss stability). *Under the notation of App. A, with $\{(B_j, B_j')\}_{j \in [J]}$ a collection of disjoint index vector pairs where $(B_j')_{j \in [J]}$ is a pairwise disjoint family, and $\rho_{n,j}(Z_{B_j'}, Z_{B_j}) \triangleq \frac{1}{|B_j'|}\sum_{i \in B_j'} h_{n,j}(Z_i, Z_{B_j})$, suppose that the datapoints $(Z_i)_{i \geq 1}$ are i.i.d. copies of a random element $Z_0$. Define $\rho_{n,j}'(Z_{B_j'}, Z_{B_j}) \triangleq \rho_{n,j}(Z_i, Z_{B_j}) - \mathbb{E}[\rho_{n,j}(Z_i, Z_{B_j}) \mid Z_{B_j}]$ and $\rho_{n,j}''(Z_{B_j'}, Z_{B_j}) \triangleq \rho_{n,j}'(Z_i, Z_{B_j}) - \mathbb{E}[\rho_{n,j}'(Z_i, Z_{B_j}) \mid Z_i]$. Then*

$$\mathbb{E}[(\tfrac{1}{J}\sum_{j=1}^{J} \rho_{n,j}''(Z_{B_j'}, Z_{B_j}))^2] \leq \tfrac{1}{J^2}\Big(\sum_{j \neq j'} \sqrt{\gamma_{loss}(h_{n,j})\gamma_{loss}(h_{n,j'})}$$
$$+ \sum_{j=1}^{J} \tfrac{1}{|B_j'|}\tfrac{1}{2}|B_j|\gamma_{loss}(h_{n,j})\Big). \tag{C.1}$$

**Proof**

Define $h_{n,j}'$ and $h_{n,j}''$ as:

$$h_{n,j}'(Z_i, Z_{B_j}) \triangleq h_{n,j}(Z_i, Z_{B_j}) - \mathbb{E}[h_{n,j}(Z_i, Z_{B_j}) \mid Z_{B_j}],$$

$$h_{n,j}''(Z_i, Z_{B_j}) \triangleq h_{n,j}'(Z_i, Z_{B_j}) - \mathbb{E}[h_{n,j}'(Z_i, Z_{B_j}) \mid Z_i].$$

Therefore, we have $\rho_{n,j}'(Z_{B_j'}, Z_{B_j}) = \frac{1}{|B_j'|}\sum_{i \in B_j'} h_{n,j}'(Z_i, Z_{B_j})$ and $\rho_{n,j}''(Z_{B_j'}, Z_{B_j}) = \frac{1}{|B_j'|}\sum_{i \in B_j'} h_{n,j}''(Z_i, Z_{B_j})$.

Thus

$$(\tfrac{1}{J}\sum_{j=1}^{J} \rho_{n,j}''(Z_{B_j'}, Z_{B_j}))^2 = \tfrac{1}{J^2}\sum_{j,j'=1}^{J} \rho_{n,j}''(Z_{B_j'}, Z_{B_j})\rho_{n,j'}''(Z_{B_{j'}'}, Z_{B_{j'}})$$
$$= \tfrac{1}{J^2}\sum_{j,j'=1}^{J} \tfrac{1}{|B_j'|}\tfrac{1}{|B_{j'}'|}\sum_{i \in B_j'}\sum_{i' \in B_{j'}'} h_{n,j}''(Z_i, Z_{B_j})h_{n,j'}''(Z_{i'}, Z_{B_{j'}}).$$

In what follows, $Z_{B_{j'}}^{\setminus i}$ is $Z_{B_{j'}}$ with $Z_i$ replaced by $Z_0'$, an i.i.d. copy of $Z_0$, independent of $(Z_i)_{i \geq 1}$. Note that if $i \notin B_{j'}$, $Z_{B_{j'}}^{\setminus i}$ is just $Z_{B_{j'}}$. We similarly define $Z_{B_j}^{\setminus i'}$.

If $j \neq j'$, we have $\mathbb{E}_{Z_i}[\sum_{i \in B'_j} \sum_{i' \in B'_{j'}} h''_{n,j}(Z_i, Z_{B_j}) h''_{n,j'}(Z_{i'}, Z_{B_{j'}}^{\setminus i})] = 0$, because (i) $h''_{n,j}(Z_i, Z_{B_j})$ and $h''_{n,j'}(Z_{i'}, Z_{B_{j'}}^{\setminus i})$ are conditionally independent given everything but $Z_i$, and (ii) $\mathbb{E}_{Z_i}[h''_{n,j}(Z_i, Z_{B_j})] = 0$.

Similarly, if $j \neq j'$,

$$\mathbb{E}_{Z_{i'}}[\sum_{i \in B'_j} \sum_{i' \in B'_{j'}} h''_{n,j}(Z_i, Z_{B_j}^{\setminus i'}) h''_{n,j'}(Z_{i'}, Z_{B_{j'}})] = 0,$$

$$\mathbb{E}_{Z_{i'}}[\sum_{i \in B'_j} \sum_{i' \in B'_{j'}} h''_{n,j}(Z_i, Z_{B_j}^{\setminus i'}) h''_{n,j'}(Z_{i'}, Z_{B_{j'}}^{\setminus i})] = 0.$$

Therefore, if $j \neq j'$,

$$\mathbb{E}[\tfrac{1}{|B'_j|}\tfrac{1}{|B'_{j'}|} \sum_{i \in B'_j} \sum_{i' \in B'_{j'}} h''_{n,j}(Z_i, Z_{B_j}) h''_{n,j'}(Z_{i'}, Z_{B_{j'}})]$$

$$= \mathbb{E}[\tfrac{1}{|B'_j|}\tfrac{1}{|B'_{j'}|} \sum_{i \in B'_j} \sum_{i' \in B'_{j'}} \big((h''_{n,j}(Z_i, Z_{B_j}) - h''_{n,j}(Z_i, Z_{B_j}^{\setminus i'}))$$
$$\times (h''_{n,j'}(Z_{i'}, Z_{B_{j'}}) - h''_{n,j'}(Z_{i'}, Z_{B_{j'}}^{\setminus i})))]$$

$$= \tfrac{1}{|B'_j|}\tfrac{1}{|B'_{j'}|} \sum_{i \in B'_j} \sum_{i' \in B'_{j'}} \mathbb{E}\big[\big((h''_{n,j}(Z_i, Z_{B_j}) - h''_{n,j}(Z_i, Z_{B_j}^{\setminus i'}))$$
$$\times (h''_{n,j'}(Z_{i'}, Z_{B_{j'}}) - h''_{n,j'}(Z_{i'}, Z_{B_{j'}}^{\setminus i})))\big]$$

$$\leq \tfrac{1}{|B'_j|}\tfrac{1}{|B'_{j'}|} \sum_{i \in B'_j} \sum_{i' \in B'_{j'}} \sqrt{\mathbb{E}\Big[(h''_{n,j}(Z_i, Z_{B_j}) - h''_{n,j}(Z_i, Z_{B_j}^{\setminus i'}))^2\Big]}$$
$$\times \sqrt{\mathbb{E}\Big[(h''_{n,j'}(Z_{i'}, Z_{B_{j'}}) - h''_{n,j'}(Z_{i'}, Z_{B_{j'}}^{\setminus i}))^2\Big]}$$

$$\leq \Big(\tfrac{1}{|B'_j|}\tfrac{1}{|B'_{j'}|} \sum_{i \in B'_j} \sum_{i' \in B'_{j'}} \mathbb{E}\Big[(h''_{n,j}(Z_i, Z_{B_j}) - h''_{n,j}(Z_i, Z_{B_j}^{\setminus i'}))^2\Big]$$
$$\times \mathbb{E}\Big[(h''_{n,j'}(Z_{i'}, Z_{B_{j'}}) - h''_{n,j'}(Z_{i'}, Z_{B_{j'}}^{\setminus i}))^2\Big]\Big)^{1/2}$$

$$= \Big(\tfrac{1}{|B'_j|}\tfrac{1}{|B'_{j'}|} \sum_{i \in B'_j} \sum_{i' \in B'_{j'}} \mathbb{E}\Big[(h''_{n,j}(Z_0, Z_{B_j}) - h''_{n,j}(Z_0, Z_{B_j}^{\setminus i'}))^2\Big]$$
$$\times \mathbb{E}\Big[(h''_{n,j'}(Z_0, Z_{B_{j'}}) - h''_{n,j'}(Z_0, Z_{B_{j'}}^{\setminus i}))^2\Big]\Big)^{1/2}$$

$$= \Big(\tfrac{1}{|B'_{j'}|} \sum_{i' \in B'_{j'}} \mathbb{E}\Big[(h''_{n,j}(Z_0, Z_{B_j}) - h''_{n,j}(Z_0, Z_{B_j}^{\setminus i'}))^2\Big]\Big)^{1/2}$$
$$\times \Big(\tfrac{1}{|B'_j|} \sum_{i \in B'_j} \mathbb{E}\Big[(h''_{n,j'}(Z_0, Z_{B_{j'}}) - h''_{n,j'}(Z_0, Z_{B_{j'}}^{\setminus i}))^2\Big]\Big)^{1/2}$$

$$= \sqrt{\gamma_{ms}(h''_{n,j})\gamma_{ms}(h''_{n,j'})} = \sqrt{\gamma_{ms}(h'_{n,j})\gamma_{ms}(h'_{n,j'})} = \sqrt{\gamma_{loss}(h_{n,j})\gamma_{loss}(h_{n,j'})},$$

where we have applied Cauchy–Schwarz inequality and Jensen's inequality, used that the datapoints are i.i.d. copies of $Z_0$ and applied the definitions of mean-square stability and loss stability.

If $j = j'$ and $i \neq i'$, then $\mathbb{E}_{Z_i}[h''_{n,j}(Z_i, Z_{B_j}) h''_{n,j}(Z_{i'}, Z_{B_j})] = 0$.

If $j = j'$ and $i = i'$, then $\mathbb{E}[h''_{n,j}(Z_i, Z_{B_j})^2] = \mathbb{E}[\mathrm{Var}(h'_{n,j}(Z_i, Z_{B_j}) \mid Z_i)]$.

We now state a conditional application of a version of the Efron–Stein inequality due to Steele [47].

**Lemma 1** (Conditional Efron–Stein inequality). *Suppose that, given $W$, the random vectors $X_{1:m}$ and $X'_{1:m}$ are conditionally independent and identically distributed and that the components of $X_{1:m}$ are conditionally independent given $W$. Then, for any suitably measurable function $f$*

$$\tfrac{1}{2}\mathbb{E}[(f(X_{1:m}, W) - f(X'_{1:m}, W))^2 \mid W] = \mathrm{Var}(f(X_{1:m}, W) \mid W)$$
$$\leq \tfrac{1}{2} \sum_{i=1}^{m} \mathbb{E}[(f(X_{1:m}, W) - f(X_{1:m}^{\setminus i}, W))^2 \mid W]$$

*where, for each $i \in [m]$, $X_{1:m}^{\setminus i}$ represents $X_{1:m}$ with $X_i$ replaced with $X'_i$.*

Using Lemma 1, we get $\mathbb{E}[\mathrm{Var}(h'_{n,j}(Z_i, Z_{B_j}) \mid Z_i)] \leq \frac{1}{2}|B_j|\gamma_{ms}(h'_{n,j}) = \frac{1}{2}|B_j|\gamma_{loss}(h_{n,j})$. Combining everything, we get

$$\mathbb{E}[(\tfrac{1}{J}\textstyle\sum_{j=1}^{J}\rho''_{n,j}(Z_{B'_j}, Z_{B_j}))^2] \leq \tfrac{1}{J^2}\Big( \textstyle\sum_{j\neq j'}\sqrt{\gamma_{loss}(h_{n,j})\gamma_{loss}(h_{n,j'})}$$
$$+ \textstyle\sum_{j=1}^{J}\tfrac{1}{|B'_j|}\tfrac{1}{2}|B_j|\gamma_{loss}(h_{n,j})\Big).$$

$\square$

In the case of $k$-fold cross-validation with equal-sized folds and i.i.d. data, the left-hand side of (C.1) becomes

$$\mathrm{Var}\Big(\tfrac{1}{n}\textstyle\sum_{j=1}^{k}\sum_{i\in B'_j}(h'_n(Z_i, Z_{B_j}) - \mathbb{E}[h'_n(Z_i, Z_{B_j}) \mid Z_i])\Big),$$

and its right-hand side simplifies to

$$\tfrac{1}{k^2}\Big(k(k-1)\sqrt{\gamma_{loss}(h_n)^2} + k\tfrac{k}{n}\tfrac{1}{2}n(1-\tfrac{1}{k})\gamma_{loss}(h_n)\Big) = \tfrac{3}{2}(1-\tfrac{1}{k})\gamma_{loss}(h_n).$$

Hence,

$$\tfrac{1}{n}\mathrm{Var}\Big(\tfrac{1}{\sqrt{n}}\textstyle\sum_{j=1}^{k}\sum_{i\in B'_j}(h'_n(Z_i, Z_{B_j}) - \mathbb{E}[h'_n(Z_i, Z_{B_j}) \mid Z_i])\Big) \leq \tfrac{3}{2}\big(1-\tfrac{1}{k}\big)\gamma_{loss}(h_n).$$

We then note that the asymptotic linearity condition (2.2) in $L^2$-norm with the choice $\bar{h}_n(z) = \mathbb{E}\big[h_n(z, Z_{1:n(1-1/k)})\big]$ can be written as

$$\tfrac{1}{\sigma_n\sqrt{n}}\textstyle\sum_{j=1}^{k}\sum_{i\in B'_j}(h'_n(Z_i, Z_{B_j}) - \mathbb{E}\big[h'_n(Z_i, Z_{B_j}) \mid Z_i\big]) \xrightarrow{L^2} 0,$$

which is implied by (3.2) when $\gamma_{loss}(h_n) = o(\sigma_n^2/n)$. Therefore, Thm. 2 follows from Thm. 7.

# D  Proof of Thm. 3: Asymptotic linearity from conditional variance convergence

Thm. 3 will follow from the following more general statement.

**Theorem 8** (Asymptotic linearity from conditional variance convergence). *Under the notation of Prop. 1, suppose that the datapoints $(Z_i)_{i\geq 1}$ are i.i.d. copies of a random element $Z_0$. If a function $\bar{h}_n$ satisfies*

$$\max(k^{q-1}, 1)\textstyle\sum_{j=1}^{k}\mathbb{E}\left[\Big(\tfrac{|B'_j|}{n\sigma_n^2}\mathrm{Var}_{Z_0}\big(h_n(Z_0, Z_{B_j}) - \bar{h}_n(Z_0)\big)\Big)^{q/2}\right] \to 0$$

*for some $q \in (0, 2]$, then $\bar{h}_n$ satisfies the $L^q$ asymptotic linearity condition (2.2). If a function $\bar{h}_n$ satisfies*

$$\textstyle\sum_{j=1}^{k}\mathbb{E}\left[\min\left(1, \tfrac{\sqrt{|B'_j|}}{\sigma_n\sqrt{n}}\sqrt{\mathrm{Var}_{Z_0}\big(h_n(Z_0, Z_{B_j}) - \bar{h}_n(Z_0)\big)}\right)\right] \to 0,$$

*then $\bar{h}_n$ satisfies the in-probability asymptotic linearity condition (2.2).*

**Proof**  In the notation of Prop. 1, for each $j \in [k]$, let

$$L_j = \tfrac{1}{|B'_j|}\textstyle\sum_{i\in B'_j}(h_n(Z_i, Z_{B_j}) - \bar{h}_n(Z_i)) - \mathbb{E}_{Z_0}[h_n(Z_0, Z_{B_j}) - \bar{h}_n(Z_0)].$$

We first note that for any non-decreasing concave $\psi$ satisfying the triangle inequality, we have

$$\mathbb{E}\left[\psi\left(\left|\frac{1}{\sigma_n\sqrt{n}}\sum_{j=1}^k|B_j'|L_j\right|\right)\right] \le \mathbb{E}\left[\psi\left(\frac{1}{\sigma_n\sqrt{n}}\sum_{j=1}^k|B_j'||L_j|\right)\right]$$

$$\le \sum_{j=1}^k \mathbb{E}\left[\psi\left(\frac{1}{\sigma_n\sqrt{n}}|B_j'||L_j|\right)\right]$$

$$= \sum_{j=1}^k \mathbb{E}\left[\mathbb{E}_{Z_{B_j'}}\left[\psi\left(\frac{1}{\sigma_n\sqrt{n}}|B_j'||L_j|\right)\right]\right]$$

$$\le \sum_{j=1}^k \mathbb{E}\left[\psi\left(\frac{1}{\sigma_n\sqrt{n}}|B_j'|\mathbb{E}_{Z_{B_j'}}[|L_j|]\right)\right]$$

$$\le \sum_{j=1}^k \mathbb{E}\left[\psi\left(\frac{1}{\sigma_n\sqrt{n}}|B_j'|\sqrt{\mathrm{Var}_{Z_{B_j'}}(L_j)}\right)\right]$$

$$= \sum_{j=1}^k \mathbb{E}\left[\psi\left(\frac{\sqrt{|B_j'|}}{\sigma_n\sqrt{n}}\sqrt{\mathrm{Var}_{Z_0}\big(h_n(Z_0, Z_{B_j}) - \bar{h}_n(Z_0)\big)}\right)\right],$$

where we have applied the triangle inequality twice, the tower property once, and Jensen's inequality twice. The advertised $L^q$ result for $q \in (0,1]$ now follows by taking $\psi(x) = x^q$, and the in-probability result follows by taking $\psi(x) = \min(1, x)$ and invoking the following lemma.

**Lemma 2.** *For any sequence of random variables $(X_n)_{n\ge1}$, $X_n \xrightarrow{p} 0$ if and only if $\mathbb{E}[\psi(|X_n|)] \to 0$, where $\psi(x) = \min(1, x)$.*

**Proof** If $X_n \xrightarrow{p} 0$, then as $X_n \xrightarrow{d} 0$ and $\psi$ is bounded and continuous for nonnegative $x$, $\mathbb{E}[\psi(|X_n|)] \to 0$. Now suppose $\mathbb{E}[\psi(|X_n|)] \to 0$. Since $\psi$ is nonnegative and non-decreasing for nonnegative $x$, we have $\mathbb{P}(|X_n| > \epsilon) \le \mathbb{E}[\psi(|X_n|)]/\psi(\epsilon) \to 0$ for every $\epsilon > 0$ by Markov's inequality. Hence, $X_n \xrightarrow{p} 0$. $\square$

Now fix any $q \in (1,2]$, and note that as $x \mapsto x^q$ is non-decreasing and convex on the nonnegative reals, we have

$$\mathbb{E}\left[\left|\frac{1}{\sigma_n\sqrt{n}}\sum_{j=1}^k|B_j'|L_j\right|^q\right] \le \mathbb{E}\left[\left(\frac{k}{k}\sum_{j=1}^k\frac{1}{\sigma_n\sqrt{n}}|B_j'||L_j|\right)^q\right]$$

$$\le \frac{k^q}{k}\sum_{j=1}^k \mathbb{E}\left[\left(\frac{1}{\sigma_n\sqrt{n}}|B_j'||L_j|\right)^q\right]$$

$$= k^{q-1}\mathbb{E}\left[\sum_{j=1}^k \mathbb{E}_{Z_{B_j'}}\left[\left(\frac{1}{n\sigma_n^2}|B_j'|^2|L_j|^2\right)^{q/2}\right]\right]$$

$$\le k^{q-1}\mathbb{E}\left[\sum_{j=1}^k\left(\frac{|B_j'|^2}{n\sigma_n^2}\mathrm{Var}_{Z_{B_j'}}(L_j)\right)^{q/2}\right]$$

$$= k^{q-1}\mathbb{E}\left[\sum_{j=1}^k\left(\frac{|B_j'|}{n\sigma_n^2}\mathrm{Var}_{Z_0}\big(h_n(Z_0, Z_{B_j}) - \bar{h}_n(Z_0)\big)\right)^{q/2}\right],$$

where we have applied the triangle inequality, Jensen's inequality using the convexity of $x \mapsto x^q$, the tower property, and Jensen's inequality using the concavity of $x \mapsto x^{q/2}$. Hence, the $L^q$ result for $q \in (1,2]$ follows from our convergence assumption. $\square$

Thm. 3 then follows from Thm. 8 by replacing $|B_j'|$ with $\frac{n}{k}$ and $Z_{B_j}$ with $Z_{1:n(1-1/k)}$ since folds are equal-sized and the $Z_i$'s are i.i.d.

# E  Conditional Variance Convergence from Loss Stability

We show that the quantity appearing in (3.3) is controlled by the loss stability, for any $q \in (0, 2]$. Note however that (3.3) can be satisfied even in a case where the loss stability is infinite (see App. G).

**Proposition 4** (Conditional variance convergence from loss stability). *Suppose that $k$ divides $n$ evenly. Under the notation of Thm. 3 with $\bar{h}_n(z) = \mathbb{E}[h_n(z, Z_{1:n(1-1/k)})]$,*

$$\mathbb{E}\left[\left(\tfrac{1}{\sigma_n^2}\mathrm{Var}_{Z_0}\big(h_n(Z_0, Z_{1:n(1-1/k)}) - \bar{h}_n(Z_0)\big)\right)^{q/2}\right] \le \left(\tfrac{1}{\sigma_n^2}\tfrac{1}{2}n(1-1/k)\gamma_{loss}(h_n)\right)^{q/2},$$

*for any $q \in (0, 2]$. Consequently, the condition (3.3) is verified whenever $\gamma_{loss}(h_n) = o\Big(\frac{\sigma_n^2}{n(1-1/k)\max(k, k^{(2/q)-1})}\Big)$.*

**Remark 2.** *If $k = O(1)$, this loss stability assumption simplifies to $\gamma_{loss}(h_n) = o(\sigma_n^2/n)$ for any $q \in (0, 2]$.*

**Proof**  Write $m = n(1 - 1/k)$. Then

$$\mathrm{Var}_{Z_0}(h_n(Z_0, Z_{1:m}) - \mathbb{E}[h_n(Z_0, Z_{1:m}) \mid Z_0]) = \mathrm{Var}_{Z_0}(h'_n(Z_0, Z_{1:m}) - \mathbb{E}[h'_n(Z_0, Z_{1:m}) \mid Z_0]),$$

since the difference $h_n(Z_0, Z_{1:m}) - h'_n(Z_0, Z_{1:m}) = \mathbb{E}[h_n(Z_0, Z_{1:m}) \mid Z_{1:m}]$ is a $Z_{1:m}$-measurable function. For $0 < q \le 2$, using Jensen's inequality,

$$\mathbb{E}\Big[(\mathrm{Var}_{Z_0}(h'_n(Z_0, Z_{1:m}) - \mathbb{E}[h'_n(Z_0, Z_{1:m}) \mid Z_0]))^{q/2}\Big]$$
$$\le \mathbb{E}[\mathrm{Var}_{Z_0}(h'_n(Z_0, Z_{1:m}) - \mathbb{E}[h'_n(Z_0, Z_{1:m}) \mid Z_0])]^{q/2}.$$

We can bound it using loss stability.

$$\mathrm{Var}_{Z_0}(h'_n(Z_0, Z_{1:m}) - \mathbb{E}[h'_n(Z_0, Z_{1:m}) \mid Z_0])$$
$$= \mathbb{E}_{Z_0}\Big[\big((h'_n(Z_0, Z_{1:m}) - \mathbb{E}[h'_n(Z_0, Z_{1:m}) \mid Z_0])$$
$$- (\mathbb{E}[h'_n(Z_0, Z_{1:m}) \mid Z_{1:m}] - \mathbb{E}[h'_n(Z_0, Z_{1:m})])\big)^2\Big]$$
$$= \mathbb{E}[(h'_n(Z_0, Z_{1:m}) - \mathbb{E}[h'_n(Z_0, Z_{1:m}) \mid Z_0])^2 \mid Z_{1:m}],$$

so that

$$\mathbb{E}[\mathrm{Var}_{Z_0}(h'_n(Z_0, Z_{1:m}) - \mathbb{E}[h'_n(Z_0, Z_{1:m}) \mid Z_0])] = \mathbb{E}[(h'_n(Z_0, Z_{1:m}) - \mathbb{E}\big[h'_n(Z_0, Z_{1:m}) \mid Z_0\big])^2]$$
$$= \mathbb{E}[\mathrm{Var}(h'_n(Z_0, Z_{1:m}) \mid Z_0)]$$
$$\le \tfrac{1}{2}m\gamma_{loss}(h_n),$$

where the last inequality comes from Lemma 1. Consequently,

$$\mathbb{E}\left[\left(\tfrac{1}{\sigma_n^2}\mathrm{Var}_{Z_0}(h_n(Z_0, Z_{1:m}) - \mathbb{E}[h_n(Z_0, Z_{1:m}) \mid Z_0])\right)^{q/2}\right] \le \left(\tfrac{1}{\sigma_n^2}\tfrac{1}{2}m\gamma_{loss}(h_n)\right)^{q/2}.$$

$\square$

# F  Excess Loss of Sample Mean: $o(\frac{\sigma_n^2}{n})$ loss stability, constant $\sigma_n^2 \in (0, \infty)$, infinite mean-square stability

Here we present a very simple learning task in which (i) the CLT conditions of Thms. 1 and 2 hold and (ii) mean-square stability (3.1) is infinite.

**Example 1** (Excess loss of sample mean: $o(\frac{\sigma_n^2}{n})$ loss stability, constant $\sigma_n^2 \in (0, \infty)$, infinite mean-square stability). *Suppose $(Z_i)_{i \ge 1}$ are independent and identically distributed copies of a random element $Z_0$ with $\mathbb{E}[Z_0] = 0$ and $\mathbb{E}[Z_0^2] < \infty$. Consider $k$-fold cross-validation of the excess loss of the sample mean relative to a constant prediction rule:*

$$h_n(z, \mathcal{D}) = (z - \hat{f}(\mathcal{D}))^2 - (z - a)^2 \quad where \quad \hat{f}(\mathcal{D}) \triangleq \tfrac{1}{|\mathcal{D}|}\sum_{Z_0 \in \mathcal{D}} Z_0 \quad and \quad a \ne 0.$$

The variance parameter of Thm. 1 $\sigma_n^2 = \mathrm{Var}(\bar{h}_n(Z_0)) = 4a^2\mathrm{Var}(Z_0)$ when $\bar{h}_n(z) = \mathbb{E}[h_n(z, Z_{1:n(1-1/k)})]$, and the loss stability $\gamma_{loss}(h_n) = \frac{8\mathrm{Var}(Z_0)^2}{n^2(1-1/k)^2} = o(\sigma_n^2/n)$. Consequently Thm. 2 implies asymptotic linearity. The uniform integrability condition of Thm. 1 also holds. Together, these results imply that the CLT of Thm. 1 is applicable. However, whenever $Z_0$ does not have a fourth moment, the mean-square stability (3.1) is infinite.

**Proof** Introduce the shorthand $m = n(1 - 1/k)$, fix any $\mathcal{D}$ with $|\mathcal{D}| = m$, and suppose $\mathcal{D}^{Z_0'}$ is formed by swapping $Z_0'$ for an independent point $Z_0''$ in $\mathcal{D}$. For any $z$ we have

$$
\begin{aligned}
(z - \hat{f}(\mathcal{D}))^2 - (z - \hat{f}(\mathcal{D}^{Z_0'}))^2 &= (\hat{f}(\mathcal{D}) - \hat{f}(\mathcal{D}^{Z_0'}))(2z - \hat{f}(\mathcal{D}) - \hat{f}(\mathcal{D}^{Z_0'})) \\
&= \tfrac{1}{m}(Z_0'' - Z_0')(2z - \hat{f}(\mathcal{D}) - \hat{f}(\mathcal{D}^{Z_0'})) \\
&= \tfrac{1}{m}(Z_0'' - Z_0')(2z - \tfrac{1}{m}(Z_0'' + Z_0') - 2(\hat{f}(\mathcal{D}) - \tfrac{1}{m}Z_0'')) \\
&= \tfrac{2}{m}(Z_0'' - Z_0')(z - (\hat{f}(\mathcal{D}) - \tfrac{1}{m}Z_0'')) - \tfrac{1}{m^2}(Z_0''^2 - Z_0'^2).
\end{aligned}
$$

Hence, the mean-square stability equals

$$
\begin{aligned}
&\mathbb{E}[((Z_0 - \hat{f}(\mathcal{D}))^2 - (Z_0 - \hat{f}(\mathcal{D}^{Z_0'}))^2)^2] \\
&= \tfrac{1}{m^4}\mathbb{E}[(Z_0''^2 - Z_0'^2)^2] + \tfrac{4}{m^2}\mathbb{E}[(Z_0'' - Z_0')^2]\mathbb{E}[(Z_0 - (\hat{f}(\mathcal{D}) - \tfrac{1}{m}Z_0''))^2] \\
&\quad - \tfrac{4}{m^3}\mathbb{E}[(Z_0'' - Z_0')(Z_0''^2 - Z_0'^2)]\mathbb{E}[Z_0 - (\hat{f}(\mathcal{D}) - \tfrac{1}{m}Z_0'')] \\
&= \tfrac{1}{m^4}\mathbb{E}[(Z_0''^2 - Z_0'^2)^2] + \tfrac{4}{m^2}\mathbb{E}[(Z_0'' - Z_0')^2]\mathbb{E}[(Z_0 - (\hat{f}(\mathcal{D}) - \tfrac{1}{m}Z_0''))^2] \\
&\geq \tfrac{2}{m^4}\mathrm{Var}(Z_0^2)
\end{aligned}
$$

since $\mathbb{E}[Z_0] = 0$, and $Z_0, Z_0', Z_0'', \hat{f}(\mathcal{D}) - \tfrac{1}{m}Z_0''$ are mutually independent.

Moreover, the loss stability equals

$$
\begin{aligned}
&\mathbb{E}[((Z_0 - \hat{f}(\mathcal{D}))^2 - (Z_0 - \hat{f}(\mathcal{D}^{Z_0'}))^2 - \mathbb{E}_{Z_0}[(Z_0 - \hat{f}(\mathcal{D}))^2 - (Z_0 - \hat{f}(\mathcal{D}^{Z_0'}))^2])^2] \\
&= \tfrac{4}{m^2}\mathbb{E}[(Z_0'' - Z_0')^2]\mathbb{E}[(Z_0 - \mathbb{E}[Z_0])^2] = \tfrac{8}{m^2}\mathrm{Var}(Z_0)^2.
\end{aligned}
$$

Finally, for any $z$, $\mathbb{E}[(z - \hat{f}(\mathcal{D}))^2 - (z - a)^2] = \tfrac{1}{m}\mathrm{Var}(Z_0) + 2za - a^2$. Consequently, for $\bar{h}_n(Z_0) = \mathbb{E}[h_n(Z_0, \mathcal{D}) \mid Z_0]$, we get the following equalities:

$$
\begin{aligned}
\sigma_n^2 &= \mathrm{Var}(\bar{h}_n(Z_0)) = 4a^2\mathrm{Var}(Z_0), \quad \text{and} \\
(\bar{h}_n(Z_0) - \mathbb{E}[\bar{h}_n(Z_0)])^2/\sigma_n^2 &= Z_0^2/\mathrm{Var}(Z_0).
\end{aligned}
$$

The distribution of $Z_0^2/\mathrm{Var}(Z_0)$ does not depend on $n$ and is integrable, so the sequence of $(\bar{h}_n(Z_0) - \mathbb{E}[\bar{h}_n(Z_0)])^2/\sigma_n^2$ is uniformly integrable. $\qquad\square$

# G  Loss of Surrogate Mean: constant $\sigma_n^2 \in (0, \infty)$, infinite $\tilde{\sigma}_n^2$, vanishing conditional variance

The following example details a simple task in which (i) the CLT conditions of Thm. 1 and Thm. 3 hold and (ii) mean-square stability, $\tilde{\sigma}_n^2$, and loss stability are infinite.

**Example 2** (Loss of surrogate mean: constant $\sigma_n^2 \in (0, \infty)$, infinite $\tilde{\sigma}_n^2$, vanishing conditional variance). *Suppose $(Z_i)_{i \geq 1}$ are independent and identically distributed copies of a random element $Z_0 = (X_0, Y_0)$ with $Z_i = (X_i, Y_i)$ and $\mathbb{E}[X_0] = \mathbb{E}[Y_0]$. Consider $k$-fold cross-validation of the following prediction rule under squared error loss:*

$$
h_n((x, y), \mathcal{D}) = (y - \hat{f}(\mathcal{D}))^2 \quad \text{where} \quad \hat{f}(\mathcal{D}) \triangleq \tfrac{1}{|\mathcal{D}|}\sum_{(X_0, Y_0) \in \mathcal{D}} X_0.
$$

*The loss stability $\gamma_{loss}(h_n) = \frac{8\mathrm{Var}(X_0)\mathrm{Var}(Y_0)}{n^2(1-1/k)^2}$, and the variance parameter of Thm. 1*

$$
\sigma_n^2 = \mathrm{Var}(\bar{h}_n(Z_0)) = \mathrm{Var}((Y_0 - \mathbb{E}[Y_0])^2), \tag{G.1}
$$

when $\bar{h}_n(z) = \mathbb{E}[h_n(z, Z_{1:n(1-1/k)})]$. Hence, if $\mathbb{E}[X_0^2], \mathbb{E}[Y_0^4] < \infty$, then $\gamma_{loss}(h_n) = o(\sigma_n^2/n)$ and Thm. 2 implies asymptotic linearity. The uniform integrability condition of Thm. 1 also holds. Together, these results imply that the CLT of Thm. 1 is applicable.

If $X_0$ has no fourth moment, then the mean-square stability (3.1) is infinite.

If $X_0$ has no second moment, then the loss stability and the [5, Theorem 1] variance parameter

$$\tilde{\sigma}_n^2 = \mathbb{E}[\text{Var}(h_n(Z_0, Z_{1:n(1-1/k)}) \mid Z_{1:n(1-1/k)})] = \text{Var}((Y_0 - \mathbb{E}[Y_0])^2) + \tfrac{8\text{Var}(X_0)\text{Var}(Y_0)}{n(1-1/k)}.$$

are infinite. However,

$$\sqrt{k}\mathbb{E}\Big[\sqrt{\tfrac{1}{\sigma_n^2}\text{Var}_{Z_0}\big(h_n(Z_0, Z_{1:n(1-1/k)}) - \bar{h}_n(Z_0)\big)}\Big]$$

$$= 2\sqrt{k}\sqrt{\tfrac{\text{Var}(Y_0)}{\text{Var}((Y_0-\mathbb{E}[Y_0])^2)}}\mathbb{E}[|\hat{f}(Z_{1:n(1-1/k)}) - \mathbb{E}[X_0]|].$$

Hence, if $\mathbb{E}[Y_0^4] < \infty$ and $k = O(1)$, $L^1$ asymptotic linearity follows from Thm. 3, the uniform integrability condition of Thm. 1 still holds, and the CLT of Thm. 1 holds with the finite variance parameter (G.1).

**Proof**    Without loss of generality, we will assume $\mathbb{E}[X_0] = \mathbb{E}[Y_0] = 0$; the formulas in the general case are obtained by replacing $X_0$ with $X_0 - \mathbb{E}[X_0]$ and similarly for $Y_0$. Introduce the shorthand $m = n(1 - 1/k)$, fix any $\mathcal{D}$ with $|\mathcal{D}| = m$, and suppose $\mathcal{D}^{Z_0'}$ is formed by swapping $Z_0$ for an independent point $Z_0''$ in $\mathcal{D}$. For any $z = (x, y)$ we have

$$(y - \hat{f}(\mathcal{D}))^2 - (y - \hat{f}(\mathcal{D}^{Z_0'}))^2 = (\hat{f}(\mathcal{D}) - \hat{f}(\mathcal{D}^{Z_0'}))(2y - \hat{f}(\mathcal{D}) - \hat{f}(\mathcal{D}^{Z_0'}))$$

$$= \tfrac{1}{m}(X_0'' - X_0')(2y - \hat{f}(\mathcal{D}) - \hat{f}(\mathcal{D}^{Z_0'}))$$

$$= \tfrac{1}{m}(X_0'' - X_0')(2y - \tfrac{1}{m}(X_0'' + X_0') - 2(\hat{f}(\mathcal{D}) - \tfrac{1}{m}X_0''))$$

$$= \tfrac{2}{m}(X_0'' - X_0')(y - (\hat{f}(\mathcal{D}) - \tfrac{1}{m}X_0'')) - \tfrac{1}{m^2}(X_0''^2 - X_0'^2).$$

Hence, the mean-square stability equals

$$\mathbb{E}[((Y_0 - \hat{f}(\mathcal{D}))^2 - (Y_0 - \hat{f}(\mathcal{D}^{Z_0'}))^2)^2]$$

$$= \tfrac{1}{m^4}\mathbb{E}[(X_0''^2 - X_0'^2)^2] + \tfrac{4}{m^2}\mathbb{E}[(X_0'' - X_0')^2]\mathbb{E}[(Y_0 - (\hat{f}(\mathcal{D}) - \tfrac{1}{m}X_0''))^2]$$

$$- \tfrac{4}{m^3}\mathbb{E}[(X_0'' - X_0')(X_0''^2 - X_0'^2)]\mathbb{E}[Y_0 - (\hat{f}(\mathcal{D}) - \tfrac{1}{m}X_0'')]$$

$$= \tfrac{1}{m^4}\mathbb{E}[(X_0''^2 - X_0'^2)^2] + \tfrac{4}{m^2}\mathbb{E}[(X_0'' - X_0')^2]\mathbb{E}[(Y_0 - (\hat{f}(\mathcal{D}) - \tfrac{1}{m}X_0''))^2]$$

$$\geq \tfrac{2}{m^4}\text{Var}((X_0 - \mathbb{E}[X_0])^2)$$

since $\mathbb{E}[Y_0] = 0$, and $Z_0, Z_0', Z_0'', \hat{f}(\mathcal{D}) - \tfrac{1}{m}X_0''$ are mutually independent.

Moreover, the loss stability equals

$$\mathbb{E}[((Y_0 - \hat{f}(\mathcal{D}))^2 - (Y_0 - \hat{f}(\mathcal{D}^{Z_0'}))^2 - \mathbb{E}_Y[(Y_0 - \hat{f}(\mathcal{D}))^2 - (Y_0 - \hat{f}(\mathcal{D}^{Z_0'}))^2])^2]$$

$$= \tfrac{4}{m^2}\mathbb{E}[(X_0'' - X_0')^2]\mathbb{E}[(Y_0 - \mathbb{E}[Y_0])^2] = \tfrac{8}{m^2}\text{Var}(X_0)\text{Var}(Y_0).$$

Next note that, for any $y, y'$,

$$\mathbb{E}[(y - \hat{f}(\mathcal{D}))^2 - (y' - \hat{f}(\mathcal{D}))^2] = (y - y')(y + y' - 2\mathbb{E}[\hat{f}(\mathcal{D})])$$

$$= (y^2 - y'^2) - 2(y - y')\mathbb{E}[\hat{f}(\mathcal{D})] = y^2 - y'^2$$

since $\mathbb{E}[X_0] = 0$. Therefore, $\text{Var}(\mathbb{E}[h_n(Z_0, \mathcal{D}) \mid Z_0]) = \tfrac{1}{2}\mathbb{E}[(Y_0^2 - Y_0'^2)^2] = \text{Var}(Y_0^2)$.

For any $y$, $\mathbb{E}[(y - \hat{f}(\mathcal{D}))^2] = y^2 + \mathbb{E}[\hat{f}(\mathcal{D})^2]$ since $\mathbb{E}[X_0] = 0$. Consequently, for $\bar{h}_n(Z_0) = \mathbb{E}[h_n(Z_0, \mathcal{D}) \mid Z_0]$, we get the following equalities:

$$\sigma_n^2 = \text{Var}(\bar{h}_n(Z_0)) = \text{Var}(Y_0^2), \quad \text{and}$$

$$(\bar{h}_n(Z_0) - \mathbb{E}[\bar{h}_n(Z_0)])^2/\sigma_n^2 = (Y_0^2 - \mathbb{E}[Y_0^2])^2/\text{Var}(Y_0^2).$$

The distribution of $(Y_0^2 - \mathbb{E}[Y_0^2])^2/\mathrm{Var}(Y_0^2)$ does not depend on $n$ and is integrable, so the sequence of $(\bar{h}_n(Z_0) - \mathbb{E}[\bar{h}_n(Z_0)])^2/\sigma_n^2$ is uniformly integrable.

Since

$$(y - \hat{f}(\mathcal{D}))^2 - (y' - \hat{f}(\mathcal{D}))^2 = (y^2 - y'^2) - 2(y - y')\hat{f}(\mathcal{D})$$

we can compute the variance parameter of [5],

$$
\begin{aligned}
\tilde{\sigma}_n^2 &= \mathbb{E}[\mathrm{Var}(h_n(Z_0, \mathcal{D}) \mid \mathcal{D})] = \mathbb{E}[(h_n(Z_0, \mathcal{D}) - \mathbb{E}[h_n(Z_0, \mathcal{D}) | \mathcal{D}])^2] \\
&= \tfrac{1}{2}\mathbb{E}[(h_n(Z_0, \mathcal{D}) - h_n(Z_0', \mathcal{D}))^2] \\
&= \tfrac{1}{2}\mathbb{E}((Y_0^2 - Y_0'^2)^2] + 4\mathbb{E}[(Y_0 - Y_0')^2]\mathbb{E}[\hat{f}(\mathcal{D})^2] - \mathbb{E}[(y^2 - y'^2)(y - y')]\mathbb{E}[\hat{f}(\mathcal{D})] \\
&= \mathrm{Var}(Y_0^2) + 8\mathrm{Var}(Y_0)\tfrac{1}{m}\mathrm{Var}(X_0),
\end{aligned}
$$

since $\mathbb{E}[\hat{f}(\mathcal{D})] = 0$ and $\hat{f}(\mathcal{D}), Y_0, Y_0'$ are mutually independent.

Finally, let's compute $\sqrt{k}\,\mathbb{E}\Big[\sqrt{\tfrac{1}{\sigma_n^2}\mathrm{Var}_{Z_0}\big(h_n(Z_0, \mathcal{D}) - \bar{h}_n(Z_0)\big)}\Big].$

For any $y, y'$,

$$
\begin{aligned}
&((y - \hat{f}(\mathcal{D}))^2 - \mathbb{E}[(y - \hat{f}(\mathcal{D}))^2]) - ((y' - \hat{f}(\mathcal{D}))^2 - \mathbb{E}[(y' - \hat{f}(\mathcal{D}))^2]) \\
&= (y^2 - y'^2) - 2(y - y')\hat{f}(\mathcal{D}) - (y^2 - y'^2) \\
&= -2(y - y')\hat{f}(\mathcal{D}),
\end{aligned}
$$

so that $\mathrm{Var}_{Z_0}\big(h_n(Z_0, \mathcal{D}) - \bar{h}_n(Z_0)\big) = \tfrac{1}{2}\mathbb{E}[(-2(Y_0 - Y_0')\hat{f}(\mathcal{D}))^2 \mid \mathcal{D}] = 4\mathrm{Var}(Y_0)(\hat{f}(\mathcal{D}))^2.$

Then

$$\sqrt{k}\mathbb{E}\Big[\sqrt{\tfrac{1}{\sigma_n^2}\mathrm{Var}_{Z_0}\big(h_n(Z_0, \mathcal{D}) - \bar{h}_n(Z_0)\big)}\Big] = 2\sqrt{k}\sqrt{\tfrac{\mathrm{Var}(Y_0)}{\mathrm{Var}(Y_0^2)}}\mathbb{E}[|\hat{f}(\mathcal{D})|]. \qquad \text{(G.2)}$$

If $\mathbb{E}[|X_0|] < \infty$, the family of empirical averages $\{\tfrac{1}{m}\sum_{i=1}^m X_i : m \geq 1\}$ is uniformly integrable and the weak law of large numbers implies that $\hat{f}(\mathcal{D})$ converges to 0 in probability. Hence, $\hat{f}(\mathcal{D}) \xrightarrow{L^1} 0$. The quantity (G.2) then goes to zero when $k = O(1)$. $\qquad \square$

# H  Proof of Prop. 2: Variance comparison

Prop. 2 will follow from the following more general result.

**Proposition 5.** *Fix any $j \in [k]$, and define $\sigma_{n,j}^2 \triangleq \mathrm{Var}(\mathbb{E}[h_n(Z_0, Z_{B_j}) \mid Z_0])$ and $\tilde{\sigma}_{n,j}^2 \triangleq \mathbb{E}[\mathrm{Var}(h_n(Z_0, Z_{B_j}) \mid Z_{B_j})]$. Then*

$$\sigma_{n,j}^2 \leq \tilde{\sigma}_{n,j}^2 \leq \sigma_{n,j}^2 + \tfrac{|B_j|}{2}\gamma_{loss}(h_n),$$

*where the first inequality is strict whenever $h_n'(Z_0, Z_{B_j}) = h_n(Z_0, Z_{B_j}) - \mathbb{E}[h_n(Z_0, Z_{B_j}) \mid Z_{B_j}]$ depends on $Z_{B_j}$.*

**Proof**   For all $j \in [k]$, we can rewrite both variance parameters.

$$
\begin{aligned}
\tilde{\sigma}_{n,j}^2 &= \mathbb{E}[\mathrm{Var}(h_n(Z_0, Z_{B_j}) \mid Z_{B_j})] \\
&= \mathbb{E}[(h_n(Z_0, Z_{B_j}) - \mathbb{E}[h_n(Z_0, Z_{B_j}) \mid Z_{B_j}])^2] \\
&= \mathbb{E}[h_n'(Z_0, Z_{B_j})^2] = \mathrm{Var}(h_n'(Z_0, Z_{B_j})).
\end{aligned}
$$

$$
\begin{aligned}
\sigma_{n,j}^2 &= \mathrm{Var}(\mathbb{E}[h_n(Z_0, Z_{B_j}) \mid Z_0]) \\
&= \mathbb{E}[(\mathbb{E}[h_n(Z_0, Z_{B_j}) \mid Z_0] - \mathbb{E}[h_n(Z_0, Z_{B_j})])^2] \\
&= \mathbb{E}[\mathbb{E}[h_n'(Z_0, Z_{B_j}) \mid Z_0]^2] = \mathrm{Var}(\mathbb{E}[h_n'(Z_0, Z_{B_j}) \mid Z_0]) \\
&= \mathrm{Var}(h_n'(Z_0, Z_{B_j})) - \mathbb{E}[\mathrm{Var}(h_n'(Z_0, Z_{B_j}) \mid Z_0)] \\
&= \tilde{\sigma}_{n,j}^2 - \mathbb{E}[\mathrm{Var}(h_n'(Z_0, Z_{B_j}) \mid Z_0)] \leq \tilde{\sigma}_{n,j}^2,
\end{aligned}
$$

where the final inequality is strict whenever $\mathbb{E}[\mathrm{Var}(h'_n(Z_0, Z_{B_j}) \mid Z_0)]$ is non-zero.

Since every non-constant variable has either infinite or strictly positive variance, $\mathbb{E}[\mathrm{Var}(h'_n(Z_0, Z_{B_j}) \mid Z_0)] = 0 \Leftrightarrow h'_n(Z_0, Z_{B_j}) = \mathbb{E}[h'_n(Z_0, Z_{B_j}) \mid Z_0]$, that is, if and only if $h'_n(Z_0, Z_{B_j}) = h_n(Z_0, Z_{B_j}) - \mathbb{E}[h_n(Z_0, Z_{B_j}) \mid Z_{B_j}]$ is independent of $Z_{B_j}$.

Finally, we know from Lemma 1 that the difference $\tilde{\sigma}^2_{n,j} - \sigma^2_{n,j} = \mathbb{E}[\mathrm{Var}(h'_n(Z_0, Z_{B_j}) \mid Z_0)] \leq \frac{1}{2}|B_j|\gamma_{loss}(h_n)$. $\qquad\square$

Prop. 2 then follows from Prop. 5 since the $Z_i$'s are i.i.d. and, when $k$ divides $n$, the only possible size for $B_j$ is $n(1 - 1/k)$.

# I  Proof of Thm. 4: Consistent within-fold estimate of asymptotic variance

We will prove the following more detailed statement from which Thm. 4 will follow.

**Theorem 9** (Consistent within-fold estimate of asymptotic variance)**.** *Suppose that* $k \leq n/2$ *and that* $k$ *divides* $n$ *evenly. Under the notation of Thm. 1 with* $m = n(1 - 1/k)$, $\bar{h}_n(z) = \mathbb{E}[h_n(z, Z_{1:m})]$, $h'_n(Z_0, Z_{1:m}) = h_n(Z_0, Z_{1:m}) - \mathbb{E}[h_n(Z_0, Z_{1:m}) \mid Z_{1:m}]$ *and* $\bar{h}'_n(z) = \mathbb{E}[h'_n(z, Z_{1:m})]$, *define the within-fold variance estimate*

$$\hat{\sigma}^2_{n,in} \triangleq \frac{1}{k}\sum_{j=1}^{k} \frac{1}{(n/k)-1} \sum_{i\in B'_j}\left(h_n(Z_i, Z_{B_j}) - \frac{k}{n}\sum_{i'\in B'_j} h_n(Z_{i'}, Z_{B_j})\right)^2.$$

*If* $(Z_i)_{i\geq 1}$ *are i.i.d. copies of a random element* $Z_0$*, then*

$$\mathbb{E}[|\hat{\sigma}^2_{n,in} - \sigma^2_n|] \leq \frac{2n^2}{n-k}\gamma_{loss}(h_n) + 2\sqrt{\frac{2n^2}{n-k}\gamma_{loss}(h_n)\sigma^2_n} + \sqrt{\frac{1}{n}\mathbb{E}[\bar{h}'_n(Z_0)^4] + \frac{3k-n}{n(n-k)}\sigma^4_n}$$

*and there exists an absolute constant* $C$ *specified in the proof such that*

$$\mathbb{E}[(\hat{\sigma}^2_{n,in} - \sigma^2_n)^2] \leq 4\frac{Cn^4}{(n-k)^2}\gamma_4(h'_n) + 8\sqrt{\frac{Cn^4}{(n-k)^2}\gamma_4(h'_n)(\frac{1}{n}\mathbb{E}[\bar{h}'_n(Z_0)^4] + \frac{3k-n}{n(n-k)}\sigma^4_n + \sigma^4_n)}$$
$$+ 2(\frac{1}{n}\mathbb{E}[\bar{h}'_n(Z_0)^4] + \frac{3k-n}{n(n-k)}\sigma^4_n) \qquad\qquad (I.1)$$

*where* $\gamma_4(h'_n) \triangleq \frac{1}{m}\sum_{i=1}^{m}\mathbb{E}[(h'_n(Z_0, Z_{1:m}) - h'_n(Z_0, Z^{\setminus i}_{1:m}))^4]$. *Here,* $Z^{\setminus i}_{1:m}$ *denotes* $Z_{1:m}$ *with* $Z_i$ *replaced by an i.i.d. copy independent of* $Z_{0:m}$.

*Moreover,*

$$\mathbb{E}[|\hat{\sigma}^2_{n,in} - \sigma^2_n|] \leq \frac{2n^2}{n-k}\gamma_{loss}(h_n) + 2\sqrt{\frac{2n^2}{n-k}\gamma_{loss}(h_n)\sigma^2_n} + \sqrt{\frac{2}{n(n/k-1)}\sigma^4_n} + o(\sigma^2_n) \qquad (I.2)$$

*whenever the sequence of* $(\bar{h}_n(Z_0) - \mathbb{E}[\bar{h}_n(Z_0)])^2/\sigma^2_n$ *is uniformly integrable.*

**Proof**

**Eliminating training set randomness**  We begin by approximating our variance estimate

$$\hat{\sigma}^2_{n,in} = \frac{1}{k}\sum_{j=1}^{k} \frac{1}{(n/k)-1} \sum_{i\in B'_j}\left(h_n(Z_i, Z_{B_j}) - \frac{k}{n}\sum_{i'\in B'_j} h_n(Z_{i'}, Z_{B_j})\right)^2$$

$$= \frac{1}{k}\sum_{j=1}^{k} \frac{1}{(n/k)-1} \sum_{i\in B'_j}\left(h'_n(Z_i, Z_{B_j}) - \frac{k}{n}\sum_{i'\in B'_j} h'_n(Z_{i'}, Z_{B_j})\right)^2$$

by a quantity eliminating training set randomness in each summand,

$$\hat{\sigma}^2_{n,in,approx} \triangleq \frac{1}{k}\sum_{j=1}^{k} \frac{1}{(n/k)-1} \sum_{i\in B'_j}\left(\bar{h}'_n(Z_i) - \frac{k}{n}\sum_{i'\in B'_j} \bar{h}'_n(Z_{i'})\right)^2,$$

where $\bar{h}'_n(z) = \mathbb{E}[h'_n(z, Z_{1:m})]$. Note that $\bar{h}'_n(Z_0)$ has expectation 0.

By Cauchy–Schwarz, we have

$$|\hat{\sigma}^2_{n,in} - \hat{\sigma}^2_{n,in,approx}| \leq \Delta + 2\sqrt{\Delta}\hat{\sigma}_{n,in,approx}$$

for the error term

$$\Delta \triangleq \frac{1}{k} \sum_{j=1}^{k} \frac{1}{(n/k)-1} \sum_{i \in B'_j} \left( h'_n(Z_i, Z_{B_j}) - \bar{h}'_n(Z_i) + \frac{k}{n} \sum_{i' \in B'_j} \bar{h}'_n(Z_{i'}) - \frac{k}{n} \sum_{i' \in B'_j} h'_n(Z_{i'}, Z_{B_j}) \right)^2$$

$$\leq 2 \frac{1}{k} \sum_{j=1}^{k} \frac{1}{(n/k)-1} \sum_{i \in B'_j} (h'_n(Z_i, Z_{B_j}) - \bar{h}'_n(Z_i))^2$$

$$+ 2 \frac{1}{k} \sum_{j=1}^{k} \frac{1}{(n/k)-1} \sum_{i \in B'_j} \left( \frac{k}{n} \sum_{i' \in B'_j} (\bar{h}'_n(Z_{i'}) - h'_n(Z_{i'}, Z_{B_j})) \right)^2$$

$$\leq \frac{2}{n-k} \sum_{j=1}^{k} \sum_{i \in B'_j} (h'_n(Z_i, Z_{B_j}) - \bar{h}'_n(Z_i))^2$$

$$+ 2 \frac{1}{k} \sum_{j=1}^{k} \frac{1}{(n/k)-1} \sum_{i \in B'_j} \frac{k}{n} \sum_{i' \in B'_j} (\bar{h}'_n(Z_{i'}) - h'_n(Z_{i'}, Z_{B_j}))^2$$

$$= \frac{4}{n-k} \sum_{j=1}^{k} \sum_{i \in B'_j} (h'_n(Z_i, Z_{B_j}) - \bar{h}'_n(Z_i))^2 \tag{I.3}$$

where we have used Jensen's inequality twice.

Thus,

$$\Delta^2 \leq \frac{16n^2}{(n-k)^2} \frac{1}{n} \sum_{j=1}^{k} \sum_{i \in B'_j} (h'_n(Z_i, Z_{B_j}) - \bar{h}'_n(Z_i))^4$$

$$= \frac{16n}{(n-k)^2} \sum_{j=1}^{k} \sum_{i \in B'_j} (h'_n(Z_i, Z_{B_j}) - \bar{h}'_n(Z_i))^4 \tag{I.4}$$

by Jensen's inequality.

**Controlling the error $\Delta$**  We will first control the error term $\Delta$. By the bound (I.3) and the conditional Efron–Stein inequality (Lemma 1), we have

$$\mathbb{E}[\Delta] \leq \frac{4}{n-k} \sum_{j=1}^{k} \sum_{i \in B'_j} \mathbb{E}[(h'_n(Z_i, Z_{B_j}) - \bar{h}'_n(Z_i))^2]$$

$$= \frac{4}{n-k} \sum_{j=1}^{k} \sum_{i \in B'_j} \mathbb{E}[(h'_n(Z_i, Z_{B_j}) - \mathbb{E}[h'_n(Z_i, Z_{B_j}) \mid Z_i])^2]$$

$$= \frac{4}{n-k} \sum_{j=1}^{k} \sum_{i \in B'_j} \mathbb{E}[\mathbb{E}[(h'_n(Z_i, Z_{B_j}) - \mathbb{E}[h'_n(Z_i, Z_{B_j}) \mid Z_i])^2 \mid Z_i]]$$

$$= \frac{4}{n-k} \sum_{j=1}^{k} \sum_{i \in B'_j} \mathbb{E}[\mathrm{Var}(h'_n(Z_i, Z_{B_j}) \mid Z_i)]$$

$$\leq \frac{2}{n-k} \sum_{j=1}^{k} \sum_{i \in B'_j} m \gamma_{ms}(h'_n)$$

$$\leq \frac{2n^2}{n-k} \gamma_{ms}(h'_n) = \frac{2n^2}{n-k} \gamma_{loss}(h_n). \tag{I.5}$$

**Controlling $\Delta^2$**  In the following, for any $j \in [k]$ and $i' \in B_j$, $Z_{B_j}^{\backslash i'}$ is $Z_{B_j}$ with $Z_{i'}$ replaced by $Z_0$. By the bound (I.4) and Boucheron et al. [12, Thm. 2], and by noting that $x^4 = x_+^4 + x_-^4$ where $x_+ = \max(x, 0)$ and $x_- = \max(-x, 0)$, we have

$$\mathbb{E}[\Delta^2] \leq \frac{16n}{(n-k)^2} \sum_{j=1}^{k} \sum_{i \in B'_j} \mathbb{E}[(h'_n(Z_i, Z_{B_j}) - \bar{h}'_n(Z_i))^4]$$

$$= \frac{16n}{(n-k)^2} \sum_{j=1}^{k} \sum_{i \in B'_j} \mathbb{E}[\mathbb{E}[(h'_n(Z_i, Z_{B_j}) - \mathbb{E}[h'_n(Z_i, Z_{B_j}) \mid Z_i])^4 \mid Z_i]]$$

$$= \frac{16n}{(n-k)^2} \sum_{j=1}^{k} \sum_{i \in B'_j} \Big( \mathbb{E}[\mathbb{E}[(h'_n(Z_i, Z_{B_j}) - \mathbb{E}[h'_n(Z_i, Z_{B_j}) \mid Z_i])_+^4 \mid Z_i]]$$

$$+ \mathbb{E}[\mathbb{E}[(h'_n(Z_i, Z_{B_j}) - \mathbb{E}[h'_n(Z_i, Z_{B_j}) \mid Z_i])_-^4 \mid Z_i]] \Big)$$

$$\leq \frac{16n}{(n-k)^2} (1 - \tfrac{1}{4})^2 4 (\tfrac{8}{7})^2 16 \sum_{j=1}^{k} \sum_{i \in B'_j} \Big( \mathbb{E}[(\mathbb{E}[\sum_{i' \in B_j} (h'_n(Z_i, Z_{B_j}) - h'_n(Z_i, Z_{B_j}^{\backslash i'}))_+^2 \mid Z_{B_j}])^2]$$

$$+ \mathbb{E}[(\mathbb{E}[\sum_{i' \in B_j} (h'_n(Z_i, Z_{B_j}) - h'_n(Z_i, Z_{B_j}^{\backslash i'}))_-^2 \mid Z_{B_j}])^2] \Big)$$

$$\leq \frac{16n}{(n-k)^2} \frac{2304}{49} \sum_{j=1}^{k} \sum_{i \in B'_j} \Big( \mathbb{E}[(\sum_{i' \in B_j} (h'_n(Z_i, Z_{B_j}) - h'_n(Z_i, Z_{B_j}^{\backslash i'}))_+^2)^2]$$

$$+ \mathbb{E}[(\sum_{i' \in B_j} (h'_n(Z_i, Z_{B_j}) - h'_n(Z_i, Z_{B_j}^{\backslash i'}))_-^2)^2] \Big)$$

$$\leq \frac{36864n}{49(n-k)^2} m \sum_{j=1}^{k} \sum_{i \in B'_j} \Big( \sum_{i' \in B_j} \mathbb{E}[(h'_n(Z_i, Z_{B_j}) - h'_n(Z_i, Z_{B_j}^{\backslash i'}))_+^4]$$

$$+ \sum_{i' \in B_j} \mathbb{E}[(h'_n(Z_i, Z_{B_j}) - h'_n(Z_i, Z_{B_j}^{\backslash i'}))_-^4] \Big)$$

$$= \tfrac{36864n}{49(n-k)^2} m \sum_{j=1}^k \sum_{i \in B'_j} \sum_{i' \in B_j} \mathbb{E}[(h'_n(Z_i, Z_{B_j}) - h'_n(Z_i, Z_{B_j}^{\setminus i'}))^4]$$

$$\leq \tfrac{36864n^4}{49(n-k)^2} \gamma_4(h'_n) = \tfrac{Cn^4}{(n-k)^2} \gamma_4(h'_n) \tag{I.6}$$

where $\gamma_4(h'_n) = \tfrac{1}{m} \sum_{i=1}^m \mathbb{E}[(h'_n(Z_0, Z_{1:m}) - h'_n(Z_0, Z_{1:m}^{\setminus i}))^4]$ and $C = 36864/49$.

**Controlling the error $\hat{\sigma}_{n,in,approx}^2 - \sigma_n^2$** To control the error $\hat{\sigma}_{n,in,approx}^2 - \sigma_n^2$, we first rewrite $\hat{\sigma}_{n,in,approx}^2$ as

$$\hat{\sigma}_{n,in,approx}^2 = \tfrac{1}{k} \sum_{j=1}^k \tfrac{1}{(n/k)-1} \sum_{i \in B'_j} \left( \bar{h}'_n(Z_i) - \tfrac{k}{n} \sum_{i' \in B'_j} \bar{h}'_n(Z_{i'}) \right)^2$$

$$= \tfrac{1}{k} \sum_{j=1}^k \tfrac{n}{n-k} \tfrac{k}{n} \sum_{i \in B'_j} \left( \bar{h}'_n(Z_i) - \tfrac{k}{n} \sum_{i' \in B'_j} \bar{h}'_n(Z_{i'}) \right)^2$$

$$= \tfrac{n}{n-k} \tfrac{1}{k} \sum_{j=1}^k \left( \tfrac{k}{n} \sum_{i \in B'_j} \bar{h}'_n(Z_i)^2 - \left( \tfrac{k}{n} \sum_{i \in B'_j} \bar{h}'_n(Z_i) \right)^2 \right)$$

$$= \tfrac{n}{n-k} \tfrac{1}{n} \sum_{j=1}^k \sum_{i \in B'_j} \bar{h}'_n(Z_i)^2 - \tfrac{n}{n-k} \tfrac{1}{k} \sum_{j=1}^k \left( \tfrac{k}{n} \sum_{i \in B'_j} \bar{h}'_n(Z_i) \right)^2.$$

We rewrite it once again to find

$$\hat{\sigma}_{n,in,approx}^2 = \tfrac{1}{n} \sum_{j=1}^k \sum_{i \in B'_j} \bar{h}'_n(Z_i)^2 - \tfrac{1}{k} \sum_{j=1}^k W_j$$

$$= \tfrac{1}{n} \sum_{i=1}^n \bar{h}'_n(Z_i)^2 - \tfrac{1}{k} \sum_{j=1}^k W_j \tag{I.7}$$

where

$$W_j \triangleq \tfrac{1}{\binom{n/k}{2}} \sum_{\substack{i,i' \in B'_j \\ i < i'}} \bar{h}'_n(Z_i)\bar{h}'_n(Z_{i'}).$$

Since $(W_j)_{j \in [k]}$ are i.i.d. with mean 0 and for $i_1 < i'_1$ and $i_2 < i'_2$

$$\mathbb{E}[\bar{h}'_n(Z_{i_1})\bar{h}'_n(Z_{i'_1})\bar{h}'_n(Z_{i_2})\bar{h}'_n(Z_{i'_2})] = 0$$

whenever $i_1 \neq i_2$ or $i'_1 \neq i'_2$, we have

$$\mathbb{E}[(\tfrac{1}{k} \sum_{j=1}^k W_j)^2] = \tfrac{1}{k}\mathrm{Var}(W_1) = \tfrac{1}{k} \tfrac{1}{\binom{n/k}{2}^2} \sum_{\substack{i,i' \in B'_j \\ i < i'}} \mathbb{E}[\bar{h}'_n(Z_i)^2 \bar{h}'_n(Z_{i'})^2]$$

$$= \tfrac{1}{k} \tfrac{1}{\binom{n/k}{2}^2} \sum_{\substack{i,i' \in B'_j \\ i < i'}} \mathbb{E}[\bar{h}'_n(Z_i)^2]\mathbb{E}[\bar{h}'_n(Z_{i'})^2]$$

$$= \tfrac{1}{k} \tfrac{1}{\binom{n/k}{2}^2} \sum_{\substack{i,i' \in B'_j \\ i < i'}} \sigma_n^4 = \tfrac{1}{k} \tfrac{1}{\binom{n/k}{2}} \sigma_n^4 = \tfrac{2}{n(n/k-1)} \sigma_n^4 \tag{I.8}$$

by noticing that $\mathbb{E}[\bar{h}'_n(Z_0)^2] = \mathrm{Var}(\bar{h}'_n(Z_0)) = \mathrm{Var}(\bar{h}_n(Z_0)) = \sigma_n^2$.

Moreover, by the independence of our datapoints, we have

$$\mathbb{E}[\bar{h}'_n(Z_{i_1})^2 \bar{h}'_n(Z_{i_2})\bar{h}'_n(Z_{i'_2})] = 0$$

for all $i_1, i_2, i'_2 \in [n]$ such that $i_2 < i'_2$, and thus

$$\mathbb{E}[(\hat{\sigma}_{n,in,approx}^2 - \sigma_n^2)^2] = \mathrm{Var}(\hat{\sigma}_{n,in,approx}^2)$$

$$= \mathrm{Var}(\tfrac{1}{n} \sum_{i=1}^n \bar{h}'_n(Z_i)^2) + \mathbb{E}[(\tfrac{1}{k} \sum_{j=1}^k W_j)^2]$$

$$= \tfrac{1}{n}\mathrm{Var}(\bar{h}'_n(Z_0)^2) + \tfrac{2}{n(n/k-1)} \sigma_n^4$$

$$= \tfrac{1}{n}\mathbb{E}[\bar{h}'_n(Z_0)^4] + \tfrac{3k-n}{n(n-k)} \sigma_n^4. \tag{I.9}$$

**Putting the pieces together** We have

$$\mathbb{E}[\sqrt{\Delta}\hat{\sigma}_{n,in,approx}] \leq \sqrt{\mathbb{E}[\Delta]\mathbb{E}[\hat{\sigma}_{n,in,approx}^2]} \leq \sqrt{\tfrac{2n^2}{n-k} \gamma_{loss}(h_n)\sigma_n^2}$$

by Cauchy–Schwarz and the bound (I.5).

We also have

$$\mathbb{E}[\Delta\hat{\sigma}_{n,in,approx}^2] \leq \sqrt{\mathbb{E}[\Delta^2]\mathbb{E}[\hat{\sigma}_{n,in,approx}^4]}$$

$$= \sqrt{\mathbb{E}[\Delta^2](\text{Var}(\hat{\sigma}_{n,in,approx}^2) + \mathbb{E}[\hat{\sigma}_{n,in,approx}^2]^2)}$$

$$\leq \sqrt{\frac{Cn^4}{(n-k)^2}\gamma_4(h_n')(\frac{1}{n}\mathbb{E}[\bar{h}_n'(Z_0)^4] + \frac{3k-n}{n(n-k)}\sigma_n^4 + \sigma_n^4)}$$

by Cauchy-Schwarz, (I.6) and (I.9).

Assembling our results with the triangle inequality and Cauchy–Schwarz for the $L^1$ bound and with Jensen's inequality for the $L^2$ bound, we find that

$$\mathbb{E}[|\hat{\sigma}_{n,in}^2 - \sigma_n^2|] \leq \mathbb{E}[|\hat{\sigma}_{n,in}^2 - \hat{\sigma}_{n,in,approx}^2|] + \mathbb{E}[|\hat{\sigma}_{n,in,approx}^2 - \sigma_n^2|]$$

$$\leq \mathbb{E}[\Delta] + 2\mathbb{E}[\sqrt{\Delta}\hat{\sigma}_{n,in,approx}] + \sqrt{\mathbb{E}[(\hat{\sigma}_{n,in,approx}^2 - \sigma_n^2)^2]}$$

$$\leq \frac{2n^2}{n-k}\gamma_{loss}(h_n) + 2\sqrt{\frac{2n^2}{n-k}\gamma_{loss}(h_n)\sigma_n^2} + \sqrt{\frac{1}{n}\mathbb{E}[\bar{h}_n'(Z_0)^4] + \frac{3k-n}{n(n-k)}\sigma_n^4}$$

and

$$\mathbb{E}[(\hat{\sigma}_{n,in}^2 - \sigma_n^2)^2] \leq 2\mathbb{E}[(\hat{\sigma}_{n,in}^2 - \hat{\sigma}_{n,in,approx}^2)^2] + 2\mathbb{E}[(\hat{\sigma}_{n,in,approx}^2 - \sigma_n^2)^2]$$

$$\leq 4\mathbb{E}[\Delta^2] + 8\mathbb{E}[\Delta\hat{\sigma}_{n,in,approx}^2] + 2\mathbb{E}[(\hat{\sigma}_{n,in,approx}^2 - \sigma_n^2)^2]$$

$$\leq 4\frac{Cn^4}{(n-k)^2}\gamma_4(h_n') + 8\sqrt{\frac{Cn^4}{(n-k)^2}\gamma_4(h_n')(\frac{1}{n}\mathbb{E}[\bar{h}_n'(Z_0)^4] + \frac{3k-n}{n(n-k)}\sigma_n^4 + \sigma_n^4)}$$

$$+ 2(\frac{1}{n}\mathbb{E}[\bar{h}_n'(Z_0)^4] + \frac{3k-n}{n(n-k)}\sigma_n^4)$$

as advertised.

In order to get the bound

$$\mathbb{E}[|\hat{\sigma}_{n,in}^2 - \sigma_n^2|] \leq \frac{2n^2}{n-k}\gamma_{loss}(h_n) + 2\sqrt{\frac{2n^2}{n-k}\gamma_{loss}(h_n)\sigma_n^2} + \sqrt{\frac{2}{n(n/k-1)}\sigma_n^4} + o(\sigma_n^2)$$

whenever the sequence of $(\bar{h}_n(Z_0) - \mathbb{E}[\bar{h}_n(Z_0)])^2/\sigma_n^2$ is uniformly integrable, i.e., the sequence of $\bar{h}_n'(Z_0)^2/\sigma_n^2$ is uniformly integrable, we need to argue that $\frac{1}{n}\sum_{i=1}^n \bar{h}_n'(Z_i)^2/\sigma_n^2 \xrightarrow{L^1} 1$. Indeed, thanks to (I.7) and (I.8), this will lead to $\mathbb{E}[|\hat{\sigma}_{n,in,approx}^2 - \sigma_n^2|] \leq \sqrt{\frac{2}{n(n/k-1)}\sigma_n^4} + o(\sigma_n^2)$.

To this end, we show that for any triangular i.i.d. array $(X_{n,i})_{n,i}$ such that $(X_{n,1})_{n\geq 1}$ is uniformly integrable, then the two conditions in the weak law of large numbers for triangular arrays of [24, Thm. 2.2.11] (stated below) are satisfied. We will also show that for such $(X_{n,i})_{n,i}$, $(S_n \triangleq \frac{1}{n}\sum_{i=1}^n X_{n,i})_{n\geq 1}$ is uniformly integrable. Together, these results will imply $L^1$ convergence. We will then choose $X_{n,i} = \bar{h}_n'(Z_i)^2/\sigma_n^2$ to get the desired result in our specific case.

**Theorem 10** (Weak law for triangular arrays [24, Thm. 2.2.11]). *For each $n$, let $X_{n,i}$, $1 \leq i \leq n$, be independent. Let $b_n > 0$ with $b_n \to \infty$, and let $\bar{X}_{n,i} = X_{n,i}\mathbb{1}[|X_{n,i}| \leq b_n]$. Suppose that as $n \to \infty$*

$$\sum_{i=1}^n \mathbb{P}(|X_{n,i}| > b_n) \to 0, \quad and \tag{I.10}$$

$$b_n^{-2}\sum_{i=1}^n \mathbb{E}[\bar{X}_{n,i}^2] \to 0. \tag{I.11}$$

*If we let $S_n = \sum_{i=1}^n X_{n,i}$ and $a_n = \sum_{i=1}^n \mathbb{E}[\bar{X}_{n,i}]$, then $(S_n - a_n)/b_n \xrightarrow{p} 0$.*

To prove our result, we specify the case of interest $b_n = n$. First, $n\mathbb{P}(|X_{n,1}| > n) \leq \mathbb{E}[|X_{n,1}|\mathbb{1}[|X_{n,1}| > n]] \leq \sup_{m\geq 1}\mathbb{E}[|X_{m,1}|\mathbb{1}[|X_{m,1}| > n]] \to 0$ as $n \to \infty$, because $(X_{n,1})_{n\geq 1}$ is uniformly integrable. Thus the first condition (I.10) holds.

Note that we then get $\mathbb{E}[X_{n,1}\mathbb{1}[|X_{n,1}| \leq n]] \to 1$ as $n \to \infty$, for our choice $X_{n,i} = \bar{h}_n'(Z_i)^2/\sigma_n^2$ which satisfies $\mathbb{E}[X_{n,i}] = 1$.

To verify the second condition (I.11), we will show that $n^{-1}\mathbb{E}[X_{n,1}^2\mathbb{1}[X_{n,1} \leq n]] \to 0$. To this end, we need the following lemma, which gives a useful formulation of uniform integrability.

**Lemma 3** (De la Vallée Poussin Theorem [41, Thm. 22]). *If $(X_n)_{n\geq 1}$ is uniformly integrable, then there exists a nonnegative increasing function $G$ such that $G(t)/t \to \infty$ as $t \to \infty$ and $\sup_n \mathbb{E}[G(X_n)] < \infty$.*

With such a function $G$, fix any $T$ such that $G(t)/t \geq 1$ for all $t \geq T$, so that $t/G(t) \leq 1$ for all $t \geq T$. Using [24, Lem. 2.2.13] for the first equality, we can write

$$
\begin{aligned}
\tfrac{1}{n}\mathbb{E}[X_{n,1}^2 \mathbb{1}[X_{n,1} \leq n]] &= \tfrac{2}{n}\int_0^\infty y\mathbb{P}(X_{n,1}\mathbb{1}[X_{n,1}\leq n] > y)dy \\
&\leq \tfrac{2}{n}\int_0^n y\mathbb{P}(X_{n,1} > y)dy \\
&= \tfrac{2}{n}(\int_0^T y\mathbb{P}(X_{n,1} > y)dy + \int_T^n y\mathbb{P}(X_{n,1} > y)dy) \\
&\leq \tfrac{T^2}{n} + \tfrac{2}{n}\int_T^n y\mathbb{P}(X_{n,1} > y)dy) \\
&\leq \tfrac{T^2}{n} + \mathbb{E}[G(X_{n,1})]\tfrac{2}{n}\int_T^n y/G(y)dy \\
&= \tfrac{T^2}{n} + o(1),
\end{aligned}
$$

where the penultimate line follows from Markov's inequality and the last line comes from the following lemma since $\sup_{y\geq T} y/G(y) \leq 1$ and $y/G(y) \to 0$.

**Lemma 4.** *If $f(y) \to 0$ as $y \to \infty$ and $\sup_{y\geq T}|f(y)| \leq M$, then $\tfrac{1}{n}\int_T^n f(y)dy \to 0$.*

**Proof**    Let $f_n(z) = f(nz)\mathbb{1}[z > T/n]$, and note that, for any $z \geq 0$, $f_n(z) \to 0$ as $n \to \infty$. Then

$$
\begin{aligned}
\tfrac{1}{n}\int_T^n f(y)dy &= \int_0^1 \mathbb{1}[z > T/n]f(nz)dz \\
&= \int_0^1 f_n(z)dz \\
&\to 0
\end{aligned}
$$

by the bounded convergence theorem.                                                                        $\square$

Consequently, the second condition (I.11) holds.

Moreover, $(S_n \triangleq \tfrac{1}{n}\sum_{i=1}^n X_{n,i})_{n\geq 1}$ is uniformly integrable whenever $(X_{n,i})_{n,i}$ is a triangular i.i.d. array such that $(X_{n,1})_{n\geq 1}$ is uniformly integrable for the following reasons:

1. $\sup_n \mathbb{E}[|S_n|] \leq \sup_n \mathbb{E}[|X_{n,1}|] < \infty$ by triangle inequality and because $(X_{n,1})_{n\geq 1}$ is uniformly integrable.

2. For any $\varepsilon > 0$, let $\delta > 0$ such that for any event $A$ satisfying $\mathbb{P}(A) \leq \delta$, $\sup_n \mathbb{E}[|X_{n,1}|\mathbb{1}[A]] \leq \varepsilon$. Such $\delta$ exists because $(X_{n,1})_{n\geq 1}$ is uniformly integrable. Then $\sup_n \mathbb{E}[|S_n|\mathbb{1}[A]] \leq \varepsilon$ by triangle inequality.

The combination of convergence in probability and uniform integrability implies convergence in $L^1$. As a result, $\tfrac{1}{n}\sum_{i=1}^n \bar{h}_n'(Z_i)^2/\sigma_n^2 \xrightarrow{L^1} 1$ as long as the sequence of $(\bar{h}_n(Z_0) - \mathbb{E}[\bar{h}_n(Z_0)])^2/\sigma_n^2 = \bar{h}_n'(Z_0)^2/\sigma_n^2$ is uniformly integrable.

Therefore, $\mathbb{E}[|\hat{\sigma}_{n,in,approx}^2/\sigma_n^2 - 1|] \leq \sqrt{\tfrac{2}{n(n/k-1)}} + o(1)$, and we get the result advertised.    $\square$

If $k \leq n/2$, which is the case here since $k < n$ and $k$ divides $n$, then $\tfrac{2}{n(n/k-1)} \to 0$ and $\tfrac{3k-n}{n(n-k)} \to 0$.

Therefore, by (I.2), we have $(\hat{\sigma}_{n,in}^2 - \sigma_n^2)/\sigma_n^2 \xrightarrow{L^1} 0$, i.e. $\hat{\sigma}_{n,in}^2/\sigma_n^2 \xrightarrow{L^1} 1$, whenever the sequence of $(\bar{h}_n(Z_0) - \mathbb{E}[\bar{h}_n(Z_0)])^2/\sigma_n^2$ is uniformly integrable and $\gamma_{loss}(h_n) = o(\tfrac{n-k}{n^2}\sigma_n^2)$, or equivalently $\gamma_{loss}(h_n) = o(\sigma_n^2/n)$ since $k \leq n/2$, and, by (I.1), we have $(\hat{\sigma}_{n,in}^2 - \sigma_n^2)/\sigma_n^2 \xrightarrow{L^2} 0$, i.e. $\hat{\sigma}_{n,in}^2/\sigma_n^2 \xrightarrow{L^2} 1$, whenever $\mathbb{E}[\bar{h}_n'(Z_0)^4] = \mathbb{E}[(\bar{h}_n(Z_0) - \mathbb{E}[\bar{h}_n(Z_0)])^4] = o(n\sigma_n^4)$ and $\gamma_4(h_n') = o(\tfrac{(n-k)^2}{n^4}\sigma_n^4)$, or equivalently $\gamma_4(h_n') = o(\sigma_n^4/n^2)$ since $k \leq n/2$.

Thm. 4 thus follows from Thm. 9.

**Strengthening of the consistency result of [5, Prop. 1]**    We provide more details about the comparison of our $L^2$-consistency result with [5, Prop. 1]. We have $\gamma_4(h_n') \leq 16\gamma_4(h_n)$ and

$\mathbb{E}[(\bar{h}_n(Z_0) - \mathbb{E}[\bar{h}_n(Z_0)])^4] \le 16\mathbb{E}[h_n(Z_0, Z_{1:m})^4]$ by Jensen's inequality. Moreover, if $\tilde{\sigma}_n^2$ converges to a non-zero constant, since $\gamma_{loss}(h_n) \le \gamma_{ms}(h_n) \le \sqrt{\gamma_4(h_n)}$, then $\gamma_{loss}(h_n) = o(\sigma_n^2/n)$ whenever $\gamma_4(h_n) = o(\sigma_n^4/n^2)$ and thus $\sigma_n^2$ converges to the same non-zero constant as $\tilde{\sigma}_n^2$ does by Prop. 2.

## J   Proof of Thm. 5: Consistent all-pairs estimate of asymptotic variance

We will prove the following more detailed statement from which Thm. 5 will follow.

**Theorem 11** (Consistent all-pairs estimate of asymptotic variance). *Suppose that $k$ divides $n$ evenly. Under the notation of Thm. 1 with $m = n(1 - 1/k)$, $\bar{h}_n(z) = \mathbb{E}[h_n(z, Z_{1:m})]$, $h'_n(Z_0, Z_{1:m}) = h_n(Z_0, Z_{1:m}) - \mathbb{E}[h_n(Z_0, Z_{1:m}) \mid Z_{1:m}]$ and $\bar{h}'_n(z) = \mathbb{E}[h'_n(z, Z_{1:m})]$, define the all-pairs variance estimate*

$$\hat{\sigma}_{n,out}^2 \triangleq \tfrac{1}{k} \sum_{j=1}^{k} \tfrac{k}{n} \sum_{i \in B'_j} \left( h_n(Z_i, Z_{B_j}) - \hat{R}_n \right)^2.$$

*If $(Z_i)_{i \ge 1}$ are i.i.d. copies of a random element $Z_0$ and $\tilde{\sigma}_n^2 = \mathbb{E}[h'_n(Z_0, Z_{1:m})^2]$, then*

$$\mathbb{E}[|\hat{\sigma}_{n,out}^2 - \sigma_n^2|] \le (1 + \tfrac{n}{k})\gamma_{ms}(h_n) + 2\sqrt{2(1 + \tfrac{n}{k})\gamma_{ms}(h_n)\tilde{\sigma}_n^2} + m\gamma_{loss}(h_n)$$
$$+ 2\sqrt{m\gamma_{loss}(h_n)(1 - \tfrac{1}{n})\sigma_n^2} + \sqrt{\tfrac{1}{n}(\mathbb{E}[\bar{h}'_n(Z_0)^4] - \sigma_n^4)} + \tfrac{1}{n}\sigma_n^2.$$

*Moreover,*

$$\mathbb{E}[|\hat{\sigma}_{n,out}^2 - \sigma_n^2|] \le (1 + \tfrac{n}{k})\gamma_{ms}(h_n) + 2\sqrt{2(1 + \tfrac{n}{k})\gamma_{ms}(h_n)\tilde{\sigma}_n^2} + m\gamma_{loss}(h_n)$$
$$+ 2\sqrt{m\gamma_{loss}(h_n)(1 - \tfrac{1}{n})\sigma_n^2} + \tfrac{1}{n}\sigma_n^2 + o(\sigma_n^2). \tag{J.1}$$

*whenever the sequence of $(\bar{h}_n(Z_0) - \mathbb{E}[\bar{h}_n(Z_0)])^2/\sigma_n^2$ is uniformly integrable.*

**Proof**

**A common training set for each validation point pair**   We begin by approximating our variance estimate

$$\hat{\sigma}_{n,out}^2 = \tfrac{1}{n} \sum_{j=1}^{k} \sum_{i \in B'_j} \left( h_n(Z_i, Z_{B_j}) - \hat{R}_n \right)^2$$
$$= \tfrac{1}{n^2} \sum_{j,j'=1}^{k} \sum_{i \in B'_j, i' \in B'_{j'}} \tfrac{1}{2}(h_n(Z_i, Z_{B_j}) - h_n(Z_{i'}, Z_{B_{j'}}))^2$$

by a quantity that employs the same training set for each pair of validation points $Z_{(i,i')}$,

$$\hat{\sigma}_{n,out,approx,1}^2 \triangleq \tfrac{1}{n^2} \sum_{j,j'=1}^{k} \sum_{i \in B'_j, i' \in B'_{j'}} \tfrac{1}{2}(h_n(Z_i, Z_{B_j}^{\setminus i'}) - h_n(Z_{i'}, Z_{B_j}^{\setminus i'}))^2$$
$$= \tfrac{1}{n^2} \sum_{j,j'=1}^{k} \sum_{i \in B'_j, i' \in B'_{j'}} \tfrac{1}{2}(h'_n(Z_i, Z_{B_j}^{\setminus i'}) - h'_n(Z_{i'}, Z_{B_j}^{\setminus i'}))^2.$$

Here, for any $j \in [k]$ and $i' \in [n]$, $Z_{B_j}^{\setminus i'}$ is $Z_{B_j}$ with $Z_{i'}$ replaced by $Z_0$. By Cauchy–Schwarz, we have

$$|\hat{\sigma}_{n,out}^2 - \hat{\sigma}_{n,out,approx,1}^2| \le \Delta_1 + 2\sqrt{\Delta_1}\hat{\sigma}_{n,out,approx,1}$$

for the error term

$$\Delta_1 \triangleq \tfrac{1}{n^2} \sum_{j,j'=1}^{k} \sum_{i \in B'_j, i' \in B'_{j'}} \tfrac{1}{2}(h_n(Z_i, Z_{B_j}) - h_n(Z_i, Z_{B_j}^{\setminus i'}) + h_n(Z_{i'}, Z_{B_j}^{\setminus i'}) - h_n(Z_{i'}, Z_{B_{j'}}))^2$$
$$\le \tfrac{1}{n^2} \sum_{j,j'=1}^{k} \sum_{i \in B'_j, i' \in B'_{j'}} (h_n(Z_i, Z_{B_j}) - h_n(Z_i, Z_{B_j}^{\setminus i'}))^2$$
$$+ \tfrac{1}{n^2} \sum_{j,j'=1}^{k} \sum_{i \in B'_j, i' \in B'_{j'}} (h_n(Z_{i'}, Z_{B_j}^{\setminus i'}) - h_n(Z_{i'}, Z_{B_{j'}}))^2, \tag{J.2}$$

where we have used Jensen's inequality in the final display.

**Controlling the error $\Delta_1$**  We will first control the error term $\Delta_1$. Note that, for $B_{j'} \neq B_j$, $|B_{j'} \backslash (B_{j'} \cap B_j)| = \frac{n}{k}$. Hence, by the bound (J.2) and the conditional Efron-Stein inequality (Lemma 1), we have

$$\mathbb{E}[\Delta_1] \leq \gamma_{ms}(h_n) + \frac{1}{n^2} \sum_{j,j'=1}^k \sum_{i \in B_j', i' \in B_{j'}'} \mathbb{E}[(h_n(Z_{i'}, Z_{B_j}^{\backslash i'}) - h_n(Z_{i'}, Z_{B_{j'}}))^2]$$

$$\leq \gamma_{ms}(h_n) + \frac{n}{k}\gamma_{ms}(h_n) = (1 + \frac{n}{k})\gamma_{ms}(h_n). \tag{J.3}$$

**Eliminating training set randomness**  We then approximate $\hat{\sigma}_{n,out,approx,1}^2$ by a quantity eliminating training set randomness in each summand,

$$\hat{\sigma}_{n,out,approx,2}^2 \triangleq \frac{1}{n^2} \sum_{j,j'=1}^k \sum_{i \in B_j', i' \in B_{j'}'} \frac{1}{2}(\bar{h}_n'(Z_i) - \bar{h}_n'(Z_{i'}))^2$$

where $\bar{h}_n'(z) = \mathbb{E}[h_n'(z, Z_{1:m})]$. Note that $\bar{h}_n'(Z_0)$ has expectation 0.

By Cauchy–Schwarz, we have

$$|\hat{\sigma}_{n,out,approx,1}^2 - \hat{\sigma}_{n,out,approx,2}^2| \leq \Delta_2 + 2\sqrt{\Delta_2}\hat{\sigma}_{n,out,approx,2}$$

for the error term

$$\Delta_2 \triangleq \frac{1}{n^2} \sum_{j,j'=1}^k \sum_{i \in B_j', i' \in B_{j'}'} \frac{1}{2}(h_n'(Z_i, Z_{B_j}^{\backslash i'}) - \bar{h}_n'(Z_i) + \bar{h}_n'(Z_{i'}) - h_n'(Z_{i'}, Z_{B_j}^{\backslash i'}))^2$$

$$\leq \frac{1}{n^2} \sum_{j,j'=1}^k \sum_{i \in B_j', i' \in B_{j'}'} (h_n'(Z_i, Z_{B_j}^{\backslash i'}) - \bar{h}_n'(Z_i))^2$$

$$+ \frac{1}{n^2} \sum_{j,j'=1}^k \sum_{i \in B_j', i' \in B_{j'}'} (\bar{h}_n'(Z_{i'}) - h_n'(Z_{i'}, Z_{B_j}^{\backslash i'}))^2, \tag{J.4}$$

where we have used Jensen's inequality in the final display.

**Controlling the error $\Delta_2$**  We will control the error term $\Delta_2$. By the bound (J.4) and the conditional Efron-Stein inequality (Lemma 1), we have

$$\mathbb{E}[\Delta_2] \leq 2\frac{m}{2}\gamma_{ms}(h_n') = m\gamma_{loss}(h_n). \tag{J.5}$$

**Controlling the error $\hat{\sigma}_{n,out,approx,2}^2 - \sigma_n^2$**  To control the error $\hat{\sigma}_{n,out,approx,2}^2 - \sigma_n^2$, we first rewrite $\hat{\sigma}_{n,out,approx,2}^2$ as

$$\hat{\sigma}_{n,out,approx,2}^2 = \frac{1}{n^2} \sum_{j,j'=1}^k \sum_{i \in B_j', i' \in B_{j'}'} \frac{1}{2}(\bar{h}_n'(Z_i) - \bar{h}_n'(Z_{i'}))^2$$

$$= \frac{1}{n^2} \sum_{i,i'=1}^n \frac{1}{2}(\bar{h}_n'(Z_i) - \bar{h}_n'(Z_{i'}))^2$$

$$= \frac{1}{n} \sum_{i=1}^n \left(\bar{h}_n'(Z_i) - \frac{1}{n}\sum_{i'=1}^n \bar{h}_n'(Z_{i'})\right)^2$$

$$= \frac{1}{n} \sum_{i=1}^n \bar{h}_n'(Z_i)^2 - \left(\frac{1}{n}\sum_{i=1}^n \bar{h}_n'(Z_i)\right)^2. \tag{J.6}$$

Since $\mathbb{E}[\bar{h}_n'(Z_i)\bar{h}_n'(Z_{i'})] = 0$ for all $i, i' \in [n]$ with $i \neq i'$ due to independence, we have

$$\mathbb{E}[\left(\frac{1}{n}\sum_{i=1}^n \bar{h}_n'(Z_i)\right)^2] = \frac{1}{n}\mathbb{E}[\bar{h}_n'(Z_0)^2] = \frac{1}{n}\sigma_n^2. \tag{J.7}$$

Furthermore,

$$\mathbb{E}[\left(\frac{1}{n}\sum_{i=1}^n \bar{h}_n'(Z_i)^2 - \sigma_n^2\right)^2] = \text{Var}(\frac{1}{n}\sum_{i=1}^n \bar{h}_n'(Z_i)^2) = \frac{1}{n}\text{Var}(\bar{h}_n'(Z_0)^2) = \frac{1}{n}(\mathbb{E}[\bar{h}_n'(Z_0)^4] - \sigma_n^4)$$

by independence. Hence,

$$\mathbb{E}[|\hat{\sigma}_{n,out,approx,2}^2 - \sigma_n^2|] \leq \sqrt{\frac{1}{n}(\mathbb{E}[\bar{h}_n'(Z_0)^4] - \sigma_n^4)} + \frac{1}{n}\sigma_n^2.$$

**Putting the pieces together**  Since each

$$\frac{1}{2}(h_n'(Z_i, Z_{B_j}^{\backslash i'}) - h_n'(Z_{i'}, Z_{B_j}^{\backslash i'}))^2 \leq h_n'(Z_i, Z_{B_j}^{\backslash i'})^2 + h_n'(Z_{i'}, Z_{B_j}^{\backslash i'})^2,$$

we have

$$\mathbb{E}[\hat{\sigma}_{n,out,approx,1}^2] \leq 2\mathbb{E}[h_n'(Z_0, Z_{1:m})^2] = 2\tilde{\sigma}_n^2$$

and hence

$$\mathbb{E}[\sqrt{\Delta_1}\hat{\sigma}_{n,out,approx,1}] \leq \sqrt{\mathbb{E}[\Delta_1]\mathbb{E}[\hat{\sigma}_{n,out,approx,1}^2]} \leq \sqrt{2(1+\tfrac{n}{k})\gamma_{ms}(h_n)\tilde{\sigma}_n^2}$$

by Cauchy–Schwarz and the bound (J.3).

Moreover, $\mathbb{E}[\hat{\sigma}_{n,out,approx,2}^2] = (1-\frac{1}{n})\sigma_n^2$, hence

$$\mathbb{E}[\sqrt{\Delta_2}\hat{\sigma}_{n,out,approx,2}] \leq \sqrt{\mathbb{E}[\Delta_2]\mathbb{E}[\hat{\sigma}_{n,out,approx,2}^2]} \leq \sqrt{m\gamma_{loss}(h_n)(1-\tfrac{1}{n})\sigma_n^2}$$

by Cauchy–Schwarz and the bound (J.5).

Assembling our results with the triangle inequality, we find that

$$
\begin{aligned}
\mathbb{E}[|\hat{\sigma}_{n,out}^2 - \sigma_n^2|] \leq {} & \mathbb{E}[|\hat{\sigma}_{n,out}^2 - \hat{\sigma}_{n,out,approx,1}^2|] + \mathbb{E}[|\hat{\sigma}_{n,out,approx,1}^2 - \hat{\sigma}_{n,out,approx,2}^2|] \\
& + \mathbb{E}[|\hat{\sigma}_{n,out,approx,2}^2 - \sigma_n^2|] \\
\leq {} & \mathbb{E}[\Delta_1] + 2\mathbb{E}[\sqrt{\Delta_1}\hat{\sigma}_{n,out,approx,1}] \\
& + \mathbb{E}[\Delta_2] + 2\mathbb{E}[\sqrt{\Delta_2}\hat{\sigma}_{n,out,approx,2}] \\
& + \sqrt{\tfrac{1}{n}(\mathbb{E}[\bar{h}_n'(Z_0)^4] - \sigma_n^4)} + \tfrac{1}{n}\sigma_n^2 \\
\leq {} & (1+\tfrac{n}{k})\gamma_{ms}(h_n) + 2\sqrt{2(1+\tfrac{n}{k})\gamma_{ms}(h_n)\tilde{\sigma}_n^2} \\
& + m\gamma_{loss}(h_n) + 2\sqrt{m\gamma_{loss}(h_n)(1-\tfrac{1}{n})\sigma_n^2} \\
& + \sqrt{\tfrac{1}{n}(\mathbb{E}[\bar{h}_n'(Z_0)^4] - \sigma_n^4)} + \tfrac{1}{n}\sigma_n^2
\end{aligned}
$$

as advertised.

We showed in the proof of Thm. 9 that $\frac{1}{n}\sum_{i=1}^n \bar{h}_n'(Z_i)^2/\sigma_n^2 \overset{L^1}{\to} 1$ whenever the sequence of $(\bar{h}_n(Z_0) - \mathbb{E}[\bar{h}_n(Z_0)])^2/\sigma_n^2 = \bar{h}_n'(Z_0)^2/\sigma_n^2$ is uniformly integrable. Thus, with (J.6) and (J.7), we get $\mathbb{E}[|\hat{\sigma}_{n,out,approx,2}^2/\sigma_n^2 - 1|] \leq 1/n + o(1)$, and the final bound advertised

$$
\begin{aligned}
\mathbb{E}[|\hat{\sigma}_{n,out}^2 - \sigma_n^2|] \leq {} & (1+\tfrac{n}{k})\gamma_{ms}(h_n) + 2\sqrt{2(1+\tfrac{n}{k})\gamma_{ms}(h_n)\tilde{\sigma}_n^2} \\
& + m\gamma_{loss}(h_n) + 2\sqrt{m\gamma_{loss}(h_n)(1-\tfrac{1}{n})\sigma_n^2} \\
& + \tfrac{1}{n}\sigma_n^2 + o(\sigma_n^2).
\end{aligned}
$$

$\square$

By the bound (J.1), $(\hat{\sigma}_{n,out}^2 - \sigma_n^2)/\sigma_n^2 \overset{L^1}{\to} 0$, i.e. $\hat{\sigma}_{n,out}^2/\sigma_n^2 \overset{L^1}{\to} 1$, if the sequence of $(\bar{h}_n(Z_0) - \mathbb{E}[\bar{h}_n(Z_0)])^2/\sigma_n^2$ is uniformly integrable, $\gamma_{loss}(h_n) = o(\sigma_n^2/n)$ and $\gamma_{ms}(h_n) = o(\min(\frac{k\sigma_n^2}{n}, \frac{k\sigma_n^4}{n\tilde{\sigma}_n^2}))$. By noticing that $\tilde{\sigma}_n^2/\sigma_n^2 \to 1$ when $\gamma_{loss}(h_n) = o(\sigma_n^2/n)$ thanks to Prop. 2, the last condition becomes $\gamma_{ms}(h_n) = o(k\sigma_n^2/n)$. Therefore, Thm. 5 follows from Thm. 11.

# K   Experimental Setup Details

Here, we provide more details about the experimental setup of Sec. 5.

## K.1   General experimental setup details

**Learning algorithms and hyperparameters**   To illustrate the performance of our confidence intervals and tests in practice, we carry out our experiments with a diverse collection of popular learning algorithms. For classification, we use the xgboost XGBRFClassifier with n_estimators=100, subsample=0.5 and max_depth=1, the scikit-learn MLPClassifier neural network with hidden_layer_sizes=(8,4,) defining the architecture and alpha=1e2, and the scikit-learn $\ell^2$-penalized LogisticRegression with solver='lbfgs' and C=1e-3. For regression, we use the xgboost XGBRFRegressor with n_estimators=100,

subsample=0.5 and max_depth=1, the scikit-learn MLPRegressor neural network with hidden_layer_sizes=(8,4,) defining the architecture and alpha=1e2, and the scikit-learn Ridge regressor with alpha=1e6. The random forest max_depth hyperparameter and neural network, logistic, and ridge $\ell^2$ regularization strengths were selected to ensure the stability of each algorithm. All remaining hyperparameters are set to their defaults, and we set random seeds for all algorithms' random states for reproducibility. We use scikit-learn [44] version 0.22.1 and xgboost [17] version 1.0.2.

**Training set sample sizes** $n$    For both datasets, we work with the following training set sample sizes $n$: 700, 1,000, 1,500, 2,300, 3,400, 5,000, 7,500, 11,000. Up to some rounding, this corresponds to a geometric sequence with growth rate 50%.

**Details on the** Higgs **dataset**    The target variable has value either 0 or 1 and there are 28 features. We initially shuffle the rows of the dataset uniformly at random and then, starting at the 5,000,001-th instance, we take 500 consecutive chunks of the largest sample size, that is 11,000. For each $n$, we take the first $n$ instances of these 500 chunks to play the role of our 500 independent replications of size $n$. The features are standardized during training in the following way: for each iteration of $k$-fold CV ($k = 10$ here), we rescale the validation fold and the remaining folds, used as training, with the mean and standard deviation of the training data. The features for the training folds then have mean 0 and variance 1.

**Details on the** FlightsDelay **dataset**    To avoid the temporal dependence issues inherent to time series datasets, we treat the complete FlightsDelay dataset as the population and thus process it differently from the Higgs dataset. For this dataset, we predict the signed log transform ($y \mapsto \text{sign}(y) \log(1 + |y|)$; this addressed the very heavy tails of $y$ on its original scale) of the delay at arrival using 4 features: the scheduled time of the journey from the origin airport to the destination airport (taxi included), the distance between the two airports, the scheduled time of departure in minutes (converted from a time to a number between 0 and 1,439) and the airline operating the plane (that we one-hot encode). We drop the instances that have missing values for at least one of these variables. Then, we perform 500 times the sampling with replacement of 11,000 points, that is the largest sample size. For each $n$, we take the first $n$ instances of these 500 chunks to play the role of our 500 independent replications of size $n$. The features are standardized during training in the same way we do for the Higgs dataset.

**Computing target test errors**    For the FlightDelays experiments, since training datapoints are sampled with replacement, the population distribution is the entirety of the FlightDelays dataset, and we use this exact population distribution to compute all test errors. For the Higgs experiments, we form a surrogate ground-truth estimate of the target test errors using the first 5,000,000 datapoints of the shuffled Higgs dataset. As an illustration, for our method where the target test error is the $k$-fold test error $R_n = \frac{1}{n} \sum_{j=1}^{10} \sum_{i \in B'_j} \mathbb{E}[h_n(Z_i, Z_{B_j}) \mid Z_{B_j}] = \frac{1}{k} \sum_{j=1}^{10} \mathbb{E}[h_n(Z_0, Z_{B_j}) \mid Z_{B_j}]$, we use these instances to compute the $k$ conditional expectations by a Monte Carlo approximation. Practically, for each training set $Z_B$, we compute the average loss on these instances of the fitted prediction rule learned on $Z_B$. Then, we evaluate the CIs and tests constructed from the 500 training sets of varying sizes $n$ sampled from the datasets.

**Random seeds**    Seeds are set in the code to ensure reproducibility. They are used for the initial random shuffling of the datasets, the sampling with replacement for the regression dataset, the random partitioning of samples in each replication, and the randomized algorithms.

## K.2   List of procedures

In our numerical experiments, we compare our procedures with the most popular alternatives from the literature. For each procedure, we give its target test error $R_n$, the estimator $\hat{R}_n$ of this target, the variance estimator $\hat{\sigma}_n^2$, the two-sided CI used in Sec. 5.1, and the one-sided test used in Sec. 5.2.

In the following, $q_\alpha$ is the $\alpha$-quantile of a standard normal distribution and $t_{\nu,\alpha}$ is the $\alpha$-quantile of a $t$ distribution with $\nu$ degrees of freedom.

1. Our 10-fold CV CLT-based test, with $\hat{\sigma}_n$ being either $\hat{\sigma}_{n,in}$ (Thm. 4) or $\hat{\sigma}_{n,out}$ (Thm. 5). The curve with $\hat{\sigma}_{n,in}$ is not displayed in our plots since the results are almost identical to those for $\hat{\sigma}_{n,out}$ and the curves are overlapping.

   - Target test error: $R_n = \frac{1}{10} \sum_{j=1}^{10} \mathbb{E}[h_n(Z_0, Z_{B_j}) \mid Z_{B_j}]$.
   - Estimator: $\hat{R}_n = \frac{1}{n} \sum_{j=1}^{10} \sum_{i \in B'_j} h_n(Z_i, Z_{B_j})$.
   - Variance estimator: $\hat{\sigma}_n^2$, either $\hat{\sigma}_{n,in}^2$ or $\hat{\sigma}_{n,out}^2$.
   - Two-sided $(1-\alpha)$-CI: $\hat{R}_n \pm q_{1-\alpha/2} \hat{\sigma}_n / \sqrt{n}$.
   - One-sided test: REJECT $H_0 \Leftrightarrow \hat{R}_n < q_\alpha \hat{\sigma}_n / \sqrt{n}$.

2. Hold-out test described, for instance, in Austern and Zhou [5, Eq. (17)].

   - Target test error: $R_n = \mathbb{E}[h_n(Z_0, Z_S) \mid Z_S]$, where $S$ is a subset of size $\lfloor n(1 - 1/10) \rfloor$ of $[n]$. Since we already have a partition for our 10-fold CV, we can use the first fold $B_1$ for $S$.
   - Estimator: $\hat{R}_n = \frac{1}{|S^c|} \sum_{i \in S^c} h_n(Z_i, Z_S)$.
   - Variance estimator: $\hat{\sigma}_n^2 = \frac{1}{|S^c|} \sum_{i \in S^c} (h_n(Z_i, Z_S) - \hat{R}_n)^2$.
   - Two-sided $(1-\alpha)$-CI: $\hat{R}_n \pm q_{1-\alpha/2} \hat{\sigma}_n \sqrt{10} / \sqrt{n}$.
   - One-sided test: REJECT $H_0 \Leftrightarrow \hat{R}_n < q_\alpha \hat{\sigma}_n \sqrt{10} / \sqrt{n}$.

3. Cross-validated $t$-test of Dietterich [22], 10 folds.

   - Target test error: $R_n = \frac{1}{10} \sum_{j=1}^{10} \mathbb{E}[h_n(Z_0, Z_{B_j}) \mid Z_{B_j}]$.
   - Estimator: $\hat{R}_n = \frac{1}{n} \sum_{j=1}^{10} \sum_{i \in B'_j} h_n(Z_i, Z_{B_j})$.
   - Variance estimator: $\hat{\sigma}_n^2 = \frac{1}{10-1} \sum_{j=1}^{10} (p_j - \hat{R}_n)^2$, where $p_j \triangleq \frac{1}{|B'_j|} \sum_{i \in B'_j} h_n(Z_i, Z_{B_j})$.
   - Two-sided $(1-\alpha)$-CI: $\hat{R}_n \pm t_{10-1,1-\alpha/2} \hat{\sigma}_n / \sqrt{10}$.
   - One-sided test: REJECT $H_0 \Leftrightarrow \hat{R}_n < t_{10-1,\alpha} \hat{\sigma}_n / \sqrt{10}$.

4. Repeated train-validation $t$-test of Nadeau and Bengio [43], 10 repetitions of 90-10 train-validation splits.

   - Target test error: $R_n = \frac{1}{10} \sum_{j=1}^{10} \mathbb{E}[h_n(Z_0, Z_{S_j}) \mid Z_{S_j}]$, where for any $j \in [10]$, $S_j$ is a subset of size $\lfloor n(1 - 1/10) \rfloor$ of $[n]$, and these 10 subsets are chosen independently.
   - Estimator: $\hat{R}_n = \frac{1}{10} \sum_{j=1}^{10} p_j$, where $p_j \triangleq \frac{1}{|S_j^c|} \sum_{i \in S_j^c} h_n(Z_i, Z_{S_j})$.
   - Variance estimator: $\hat{\sigma}_n^2 = \frac{1}{10-1} \sum_{j=1}^{10} (p_j - \hat{R}_n)^2$.
   - Two-sided $(1-\alpha)$-CI: $\hat{R}_n \pm t_{10-1,1-\alpha/2} \hat{\sigma}_n / \sqrt{10}$.
   - One-sided test: REJECT $H_0 \Leftrightarrow \hat{R}_n < t_{10-1,\alpha} \hat{\sigma}_n / \sqrt{10}$.

5. Corrected repeated train-validation $t$-test of Nadeau and Bengio [43], 10 repetitions of 90-10 train-validation splits.

   - Target test error: $R_n = \frac{1}{10} \sum_{j=1}^{10} \mathbb{E}[h_n(Z_0, Z_{S_j}) \mid Z_{S_j}]$, where for any $j \in [10]$, $S_j$ is the same as in the previous procedure.
   - Estimator: $\hat{R}_n = \frac{1}{10} \sum_{j=1}^{10} p_j$, where $p_j$ is the same as in the previous procedure.
   - Variance estimator: $\hat{\sigma}_n^2 = (\frac{1}{10} + \frac{0.1}{1-0.1}) \frac{10}{10-1} \sum_{j=1}^{10} (p_j - \hat{R}_n)^2$.
   - Two-sided $(1-\alpha)$-CI: $\hat{R}_n \pm t_{10-1,1-\alpha/2} \hat{\sigma}_n / \sqrt{10}$.
   - One-sided test: REJECT $H_0 \Leftrightarrow \hat{R}_n < t_{10-1,\alpha} \hat{\sigma}_n / \sqrt{10}$.

6. $5 \times 2$-fold CV test of Dietterich [22].

   - Target test error: $R_n = \frac{1}{5} \sum_{j=1}^{5} \frac{1}{2} (\mathbb{E}[h_n(Z_0, Z_{B_{1,j}}) \mid Z_{B_{1,j}}] + \mathbb{E}[h_n(Z_0, Z_{B_{2,j}}) \mid Z_{B_{2,j}}])$, where for any $j \in [5]$, $\{B_{1,j}^c, B_{2,j}^c\}$ is a partition of $[n]$ into 2 folds of size $n/2$, and these 5 partitions are chosen independently.
   - Estimator: $\hat{R}_n = \frac{1}{|B_{1,1}^c|} \sum_{i \in B_{1,1}^c} h_n(Z_i, Z_{B_{1,1}})$.

- Variance estimator: $\hat{\sigma}_n^2 = \frac{1}{5}\sum_{j=1}^5 s_j^2$, where $s_j^2 \triangleq (p_{1,j} - \bar{p}_j)^2 + (p_{2,j} - \bar{p}_j)^2$ with $\bar{p}_j \triangleq (p_{1,j} + p_{2,j})/2$ and $p_{k,j} \triangleq \frac{1}{|B_{k,j}^c|}\sum_{i\in B_{k,j}^c} h_n(Z_i, Z_{B_{k,j}})$ for $k \in [2], j \in [5]$.

- Two-sided $(1-\alpha)$-CI: $\hat{R}_n \pm t_{5,1-\alpha/2}\hat{\sigma}_n$.

- One-sided test: REJECT $H_0 \Leftrightarrow \hat{R}_n < t_{5,\alpha}\hat{\sigma}_n$.

### K.3 Concentration-based confidence intervals

For comparison in Sec. 1, we also implemented the ridge regression CI from [16, Thm. 3] for the
`FlightDelays` experiment (an implementable CI is not provided for any other learning algorithm
in [16]). This CI takes as input a uniform bound $B_Y$ on the absolute value of the target variable
$Y$ and a uniform bound $B_X$ on the $\ell^2$ norm of the feature vector $X$. After mean-centering, we
find the maximum absolute value of $Y$ across the `FlightDelays` dataset to be $B_Y = 8.03$. After
mean-centering, we find the maximum $\ell^2$ norm of a feature vector $X$ across the `FlightDelays` to
be $B_X = 13.17$ if each feature is normalized to have standard deviation 1 or $B_X = 4200$ if the
features are left unnormalized. When normalizing features as in Fig. 5, the smallest width produced
by [16, Thm. 3] for any value of $n$ is 90.2; that is 91 times larger than the largest width of our CLT
intervals (equal to 0.99). When not normalizing as in Fig. 3, our maximum width is 0.98, but the
minimum [16, Thm. 3] width is $5 \times 10^{14}$.

### K.4 Leave-one-out cross-validation

To evaluate the LOOCV CLT-based CIs discussed in Sec. 5.4 we follow the ridge regression ex-
perimental setup of App. K.1 except that we regress onto the raw feature values instead of the
standardized features values described in App. K.1. For our LOOCV CLT-based CIs, the quantities
of interest are the following.

- Target test error: $R_n = \frac{1}{n}\sum_{i=1}^n \mathbb{E}[h_n(Z_0, Z_{\{i\}^c}) \mid Z_{\{i\}^c}]$.

- Estimator: $\hat{R}_n = \frac{1}{n}\sum_{i=1}^n h_n(Z_i, Z_{\{i\}^c})$ computed efficiently using the Sherman–
  Morrison–Woodbury derivation below.

- Variance estimator: $\hat{\sigma}_{n,out}^2$ with $k = n$ folds.

- Two-sided $(1-\alpha)$-CI: $\hat{R}_n \pm q_{1-\alpha/2}\hat{\sigma}_{n,out}/\sqrt{n}$.

**Results** We construct $95\%$ CIs for ridge regression test error based on our LOOCV CLT and
compare their coverage and width with those of the procedures described in Sec. 5.1. We see that,
like the 10-fold CV CLT intervals, the LOOCV intervals provide coverage near the nominal level
and widths smaller than the popular alternatives from the literature; in fact, the 10-fold CV CLT
curves are obscured by the nearly identical LOOCV CLT curves.

Figure 3: Test error coverage (left) and width (right) of $95\%$ confidence intervals for ridge regres-
sion, including leave-one-out CV intervals (see Sec. 5.4). The CV CLT curves are obscured by the
nearly identical LOOCV CLT curves.

**Efficient computation** We explain here how the Sherman–Morrison–Woodbury formula can be used to efficiently compute the individual losses $h_n(Z_i, Z_{\{i\}^c})$, and therefore $\hat{R}_n$ as well as $\hat{\sigma}_{n,out}$, and the loss on the instances used to form a surrogate ground-truth estimate of the target error $R_n$. Let $X \in \mathbb{R}^{n \times p}$ be the matrix of predictors, whose $i$-th row is $x_i^\top$, and $Y \in \mathbb{R}^n$ be the target variable. The weight vector estimate $\hat{w}$ minimizes $\min_{w \in \mathbb{R}^p} \|Y - Xw\|_2^2 + \lambda \|w\|_2^2$, and is given by the closed-form formula

$$\hat{w} = (X^\top X + \lambda I_p)^{-1} X^\top Y.$$

We precompute $M \triangleq (X^\top X + \lambda I_p)^{-1}$ and $v \triangleq X^\top Y$, that satisfy $\hat{w} = Mv$. Suppose that we have an additional set with covariate matrix $\tilde{X}$ and target variable $\tilde{Y}$, representing the instances used to form a surrogate ground-truth estimate of $R_n$. We also precompute $q \triangleq \tilde{X}\hat{w}$ and $A \triangleq \tilde{X}M$.

For the datapoint $i$, let $X^{(-i)}$ denote $X$ without its $i$-th row and $Y^{(-i)}$ denote $Y$ without its $i$-th element. Let $M_i \triangleq (X^{(-i)^\top} X^{(-i)} + \lambda I_p)^{-1}$, $v_i \triangleq X^{(-i)^\top} Y^{(-i)}$ and $w_i \triangleq M_i v_i$. We can efficiently compute $M_i$ from $M$ based on the Sherman–Morrison–Woodbury formula.

$$\begin{aligned} M_i &= (X^{(-i)^\top} X^{(-i)} + \lambda I_p)^{-1} \\ &= (X^\top X - x_i x_i^\top + \lambda I_p)^{-1} \\ &= M - M x_i x_i^\top M / (-1 + h_i), \end{aligned}$$

where $h_i \triangleq x_i^\top M x_i$.

We can compute $v_i$ from $v$, with $v_i = X^{(-i)^\top} Y^{(-i)} = v - x_i y_i$.

Therefore, $w_i = M_i v_i = (M - M x_i x_i^\top M (-1 + h_i)^{-1})(v - x_i y_i) = \hat{w} + M x_i(\langle \hat{w}, x_i \rangle - y_i)/(1 - h_i)$ can be computed without fitting any additional prediction rule. Then $h_n(Z_i, Z_{\{i\}^c}) = (y_i - \langle w_i, x_i \rangle)^2$, and we use them to compute $\hat{R}_n$ and $\hat{\sigma}_{n,out}$. To make predictions for the covariate matrix $\tilde{X}$, we efficiently compute $\tilde{X} w_i$ as

$$\begin{aligned} \tilde{X} w_i &= \tilde{X}\hat{w} + \tilde{X} M x_i(\langle \hat{w}, x_i \rangle - y_i)/(1 - h_i) \\ &= q + A x_i(\langle \hat{w}, x_i \rangle - y_i)/(1 - h_i), \end{aligned}$$

and $\frac{1}{N}\|\tilde{Y} - \tilde{X} w_i\|_2^2$ is an estimate of $\mathbb{E}[h_n(Z_0, Z_{\{i\}^c}) \mid Z_{\{i\}^c}]$, where $N$ is the size of the whole dataset. An estimate of $R_n$ is then $\frac{1}{n}\sum_{i=1}^n \frac{1}{N}\|\tilde{Y} - \tilde{X} w_i\|_2^2$.

# L  Additional Experimental Results

This section reports the additional results of the experiments described in Sec. 5.

## L.1  Additional results from Sec. 5.1: Confidence intervals for test error

The remaining results of the experiments described in Sec. 5.1 are provided in Figs. 4 and 5. We remind that each mean width estimate is displayed with a $\pm 2$ standard error confidence band, while the confidence band surrounding each coverage estimate is a 95% Wilson interval. For all 6 learning tasks, all procedures except the repeated train-validation $t$ interval provide near-nominal coverage, and our CV CLT intervals provide the smallest widths.

## L.2  Additional results from Sec. 5.2: Testing for improved algorithm performance

In this section, we provide additional experimental details and results for the testing for improved algorithm performance experiments of Sec. 5.2. We highlight that the aim of this assessment is not to establish power convergence or to assess power in an absolute sense but rather to verify whether, for a diversity of settings encountered in real learning problems, our proposed tests provide power comparable to or better than the most popular heuristics from the literature. For all testing experiments, we estimate size as $\frac{\text{\# of rejections in } H_0 \text{ replications}}{\text{\# } H_0 \text{ replications}}$ and power as $\frac{\text{\# of rejections in } H_1 \text{ replications}}{\text{\# } H_1 \text{ replications}}$, where each simulation is classified as $H_0$ or $H_1$ depending on which algorithm has smaller test

Figure 4: Test error coverage (left) and width (right) of $95\%$ confidence intervals (see Sec. 5.1). **Top:** $\ell^2$-regularized logistic regression classifier. **Middle:** Random forest classifier. **Bottom:** Neural network classifier.

error. Moreover, a size point is only displayed if at least 25 replications were classified as $H_0$, and a power point is only displayed if at least 25 replications were classified as $H_1$.

The remaining results of the testing experiments described in Sec. 5.2 are provided in Figs. 6 to 11. In contrast to Fig. 2,[9] in Figs. 6 to 11 we plot the size and power of the level $\alpha = 0.05$ test of $H_1 : \mathrm{Err}(\mathcal{A}_1) < \mathrm{Err}(\mathcal{A}_2)$ in the left column of each figure and the size and power of the level $\alpha$ test of $H_1 : \mathrm{Err}(\mathcal{A}_2) < \mathrm{Err}(\mathcal{A}_1)$ in the right column. Notably, we only observe size estimates exceeding the level when the number of $H_0$ replications is very small (that is, when one algorithm improves upon the other so infrequently that the Monte Carlo error in the size estimate is large).

## L.3  Testing with synthetically generated labels

We complement the real-data hypothesis testing experiments of Sec. 5.2 with a controlled experiment in which class labels are synthetically generated from a known logistic regression distribution. Specifically, we replicate the exact classification experimental setup of Sec. 5.2 to compare logistic regression and random forest classification and use the same `Higgs` dataset covariates, but we replace each datapoint label $Y_i$ with an independent draw from the logistic regression distribution $Y_i \sim \mathrm{Ber}(\frac{1}{1+\exp(-\langle X_i, \beta \rangle)})$ for $\beta$ a 28-dimensional vector with odd entries equal to 1 and even en-

Figure 5: Test error coverage (left) and width (right) of $95\%$ confidence intervals (see Sec. 5.1). **Top:** Random forest regression. **Middle:** Ridge regression. **Bottom:** Neural network regression.

tries equal to $-1$. This experiment enables us to evaluate our hypothesis tests in a *realizable* setting in which the true label generating distribution belongs to the logistic regression model family. In Fig. 12, we plot in the left column the size and power of the level $\alpha = 0.05$ test of $H_1$: random forest improves upon $\ell^2$-regularized logistic regression classifier, and in the right column the size and power of the level $\alpha$ test of $H_1$: $\ell^2$-regularized logistic regression classifier improves upon random forest. As expected, almost all replications satisfy that $\ell^2$-regularized logistic regression improves upon random forest and we observe that in this setting as well, our method consistently outperforms other alternatives.

## L.4 Results for Sec. 5.3: Importance of stability

In this section, we provide the figures (Figs. 13 to 15) and experimental details supporting the importance of stability experiment of Sec. 5.3. Compared to the chosen hyperparameters described in App. K, for this example, we used the default value of `max_depth` for `XGBRFRegressor`, that is 6, and the default value of `alpha` for `MLPRegressor`, that is 1e-4. For Figs. 15a and 15b, we obtain an estimate of $\sigma_n^2 = \mathrm{Var}(\bar{h}_n(Z_0))$ by computing a Monte Carlo approximation of $\bar{h}_n(Z_0) = \mathbb{E}[h_n(Z_0, Z_{1:m}) \mid Z_0]$ for each of 10,000 $Z_0$ values and then reporting the empirical variance of these 10,000 approximated values. For each value of $Z_0$ we employ the Monte Carlo approximation of

$$\bar{h}_n(Z_0) \approx \frac{1}{500} \sum_{\ell=1}^{500} \frac{1}{k} \sum_{j=1}^{k} h_n(Z_0, Z_{B_j}^{(\ell)})$$

Figure 6: Size (top) and power (bottom) of level-0.05 tests for improved test error (see Sec. 5.2). **Left**: Testing $H_1$: neural network improves upon $\ell^2$-regularized logistic regression classifier. **Right**: Testing $H_1$: $\ell^2$-regularized logistic regression classifier improves upon neural network.

Figure 7: Size (top) and power (bottom) of level-0.05 tests for improved test error (see Sec. 5.2). **Left**: Testing $H_1$: $\ell^2$-regularized logistic regression classifier improves upon random forest. **Right**: Testing $H_1$: random forest improves upon $\ell^2$-regularized logistic regression classifier.

where $(Z_{1:n}^{(\ell)})_{\ell=1}^{500}$ are the 500 datasets of size $n$ described in App. K.

Figure 8: Size (top) and power (bottom) of level-0.05 tests for improved test error (see Sec. 5.2). **Left**: Testing $H_1$: neural network classifier improves upon random forest. **Right**: Testing $H_1$: random forest classifier improves upon neural network.

Figure 9: Size (top) and power (bottom) of level-0.05 tests for improved test error (see Sec. 5.2). **Left**: Testing $H_1$: ridge regression improves upon random forest. **Right**: Testing $H_1$: random forest improves upon ridge regression.

Figure 10: Size (top) and power (bottom) of level-0.05 tests for improved test error (see Sec. 5.2).
**Left**: Testing $H_1$: neural network improves upon ridge regression. **Right**: Testing $H_1$: ridge regression improves upon neural network.

Figure 11: Size (top) and power (bottom) of level-0.05 tests for improved test error (see Sec. 5.2).
**Left**: Testing $H_1$: neural network regression improves upon random forest. **Right**: Testing $H_1$: random forest regression improves upon neural network.

Figure 12: Size (top) and power (bottom) of level-0.05 tests for improved test error **with synthetic logistic regression labels** (see App. L.3). **Left**: Testing $H_1$: random forest improves upon $\ell^2$-regularized logistic regression classifier. **Right**: Testing $H_1$: $\ell^2$-regularized logistic regression classifier improves upon random forest.

Figure 13: **Impact of instability** on size (left) and power (right) of level-0.05 tests for improved test error (see Sec. 5.3). Testing $H_1$: less stable neural network regression improves upon less stable random forest.

Figure 14: **Impact of instability** on test error coverage (top) and width (bottom) of $95\%$ confidence intervals (see Sec. 5.3). **Left:** Less stable neural network regression. **Right:** Less stable random forest regression.

(a) Algorithm comparison

(b) Single algorithm assessment

Figure 15: **Impact of instability** on variance of $\frac{\sqrt{n}}{\sigma_n}(\hat{R}_n - R_n)$ (see Sec. 5.3). **Left:** $h_n(Z_0, Z_B) = (Y_0 - \hat{f}_1(X_0; Z_B))^2 - (Y_0 - \hat{f}_2(X_0; Z_B))^2$ for neural network and random forest prediction rules, $\hat{f}_1$ and $\hat{f}_2$. As predicted in Thms. 1 and 2, the variance is close to 1 when $h_n$ is stable, but the variance can be much larger when $h_n$ is unstable. **Right:** $h_n(Z_0, Z_B) = (Y_0 - \hat{f}(X_0; Z_B))^2$ for neural network or random forest prediction rule, $\hat{f}$. The same destabilized algorithms produce relatively stable $h_n$ in the context of single algorithm assessment, as the variance parameter $\sigma_n^2 = \text{Var}(\bar{h}_n(Z_0))$ is larger.

## Footnotes

[9]Recall that in Fig. 2 we identified the algorithm $\mathcal{A}_1$ that more often had smaller test error across our simulations and displayed the power of $H_1 : \mathrm{Err}(\mathcal{A}_1) < \mathrm{Err}(\mathcal{A}_2)$ and the size of the level $\alpha = 0.05$ test of $H_1 : \mathrm{Err}(\mathcal{A}_2) < \mathrm{Err}(\mathcal{A}_1)$.