[Reviews · NeurIPS 2020]

Review 1

Summary and Contributions: This paper considers the problem of inference on cross validation error, where we aim to take into account the randomness in calculating the cross validation error. The idea in the paper is to re-format the residual between \hat R_n - R_n (estimated minus true error) in terms of another term that is more amenable to normality analysis and a "linearity condition". The authors then make an interesting connection of the linearity condition with work in algorithmic stability, and show when the condition holds. These results imply asymptotic normality for the aforementioned residual, which can also be used for model selection. The idea is illustrated through several experiments.

Strengths: Variance of CV error is important to quantify and take into account for machine learning.

Weaknesses: Some major comments: 1) The connection to algorithmic stability is interesting, but I am not convinced that this can deliver as strong results as we would like beyond what can already be achieved through standard results/analysis. More specifically, algorithmic stability has mostly shown O(1/n) results for ERM or SGD, but this is just a rehashing of standard results, essentially following from iid-ness, that is, that every datapoint contributes the same information on average. This is not a problem with the current paper per se, but more a critique of algorithmic stability analysis. Rather, my concern for the current paper is twofold: a) the connection to algorithmic stability cannot deliver, as far as I understand, any stronger results than what is already possible through standard methods; b) and thus a basic CLT for CV error is attainable through a more standard analysis. Indeed, the path to asymptotic normality is pretty straightforward in the paper, since all important steps are more-or-less assumed: Square integrability of mean loss \bar h_n, song convexity of such loss function which guarantees O(1/n) rates, etc. 2) The experimental setup is very confusing to me. I would suggest a complete rewrite. A couple of thoughts: a) This section should start with a simple example where R_n is known, instead of having to estimate it. Perhaps a linear regression model? I understand that there is an issue of space, but earlier sections could be shortened. b) Most CV methods are doing fine in Fig 1 in terms of coverage. There is a consistent finding that the method in the paper gives shorter CI widths. However, this is neither mentioned in the main paper as a strength of the method, nor is it theoretically anticipated. So, it's hard to appreciate it. c) I am not sure what to make of Fig 2. All CV tests are severely distorted, which is probably because of the simulation setup. One problem with the setup is that the "goalpost is moving"; that is, the power function is not constant as the best method changes mid-simulation. Wouldn't be much cleaner to focus on a case where ground-truth is known (e.g., normal model), and we use A1=true model, A2=misspecified model? Then, we could slowly introduce each additional complexity and illustrate how this affects the power plots? Minor comments: 1) I don't like the term "asymptotically exact". A method is either exact or not. "Asymptotically valid" is preferable. 2) \hat R_n and R_n are defined conditionally on the partition V_n. How does this affect the analysis, particularly the rate of convergence to normality? 3) what is Var_Z_0 in Equation (3.3)?

Correctness: They seem to be correct. The empirical methodology is unclear, especially its performance.

Clarity: Yes. It is very well written.

Relation to Prior Work: Yes, prior work is discussed thoroughly.

Reproducibility: Yes

Additional Feedback:


Review 2

Summary and Contributions: This work develops central limit theorems for cross-validation and consistent estimators of the asymptotic variance under weak stability conditions on the learning algorithm.

Strengths: This work develops central limit theorems for cross-validation and consistent estimators of the asymptotic variance under weak stability conditions on the learning algorithm.

Weaknesses: None

Correctness: yes

Clarity: yes

Relation to Prior Work: yes

Reproducibility: Yes

Additional Feedback:


Review 3

Summary and Contributions: The paper is devoted to cross-validation estimation of the error rate, in statistical learning problems, where the risk take the form of the expectation of an integrable r.v. and learning is based on i.i.d. data. The main result is a pivotal CLT for the standardised version of the k-fold CV estimator, stated in Theorem 1, under the assumption of asymptotic linearity, characterised in Proposition 1. Linearization (e.g. delta method, Hajek projection) being the most common tool for proving a CLT for statistics more complex than basic i.i.d. averages, the contribution of the article essentially consists in exhibiting sufficient conditions for the asymptotic linearity condition to hold true. In Theorem 2, loss stability is shown to imply asymptotic linearity, whereas Theorem 3 provides weaker sufficient condition, when k stays bounded in particular. It is proposed to build asymptotic confidence intervals based on the CLT established, while illustrative numerical experiments are detailed in section 5. Proofs, additional technical details, and a description of the experimental setup are deferred to the Supplementary Material.

Strengths: The article copes with an important topic in statistical learning, crucial in risk assessment and model selection. The conditions exhibited in the paper seems to be novel and compare favourably with the state-of-the-art. Theoretical results are nicely illustrated by numerical experiments.

Weaknesses: A possible reproach may lie in the asymptotic nature of the analysis carried out, whereas guarantees in statistical learning are usually non asymptotic.Non asymptotic bounds for CV estimates have been reviewed for instance in Cornec (2012) It would have been interesting to explain why non asymptotic linearity is possibly difficult to guarantee. In addition, placing ourselves from the asymptotic perspective, it is unclear why asymptotic unbiasedness is so desirable. Why is a minimisation of AMSE not more appropriate? This could have the advantage to relax the constraints on k and offer more flexibility in the estimation.

Correctness: I have read the proofs as carefully as I could and found no mistake, the analysis seems correct to me.

Clarity: The paper is well structured and written with clarity.

Relation to Prior Work: The state-of-the-art seems exhaustive when adopting an asymptotic viewpoint, it should be completed by mentioning non asymptotic analyses.

Reproducibility: Yes

Additional Feedback:


Review 4

Summary and Contributions: The paper derives asymptotically exact confidence bounds for the popular k-fold cross validation technique. Weaker assumptions are made relative to previous work. In particular, bounds are derived using an abstract property called asymptotic linearity. Experiments are conducted on real datasets.

Strengths: 1) Popular validation procedure analyzed 2) Bounds are asymptotically tight and based on weaker assumptions 3) Sufficient intuitive conditions are provided for their core assumption 4) Real data experiments are conducted

Weaknesses: 1) More discussion about how intuitively their assumptions are weaker than related work would have been good 2) Presentation could be improved

Correctness: Yes, as far as I checked

Clarity: Somewhat. Presentation could be improved but is reasonable.

Relation to Prior Work: Yes, although more discussion about how intuitively their assumptions are weaker than related work would have been good

Reproducibility: Yes

Additional Feedback: The paper derives asymptotically exact confidence bounds for the popular k-fold cross validation technique. Weaker assumptions are made relative to previous work. In particular, bounds are derived using an abstract property called asymptotic linearity. Experiments are conducted on real datasets. Post rebuttal ----------------- I have read the author response and my opinion is unchanged. The contribution seems to be significant given the weaker conditions which are weaker than loss stability as it implies their condition. I like that the authors took the effort to provide more interpretable sufficient conditions for their abstract assumption of asymptotic linearity rather than just proving more general results. Also conducting experiments using real data is a plus as it shows that their estimator is practical. I would have liked more discussion about how intuitively their assumptions are weaker than related work, although some discussion is provided in section 3.3. In my opinion its interesting work, although I am not an expert in judging how much of a leap the results truly represent.

[Author Response · NeurIPS 2020]

We thank all reviewers for their time and feedback; we address common and individual comments in turn.

**(R3, R1) Non-asymptotic intervals, improved widths:** In the revision we will highlight that non-asymptotic CIs can
be derived from the CV concentration inequalities of [10,11,2,3,Cornec arXiv:1011.5133]. These CIs are more difficult
to deploy as they require (1) stronger stability than loss stability, (2) a known upper bound on stability, and (3) either a
known upper bound on the loss or a known uniform bound on the covariates and a known sub-Gaussianity constant for
the response variable. In addition, the reliance on somewhat loose inequalities typically leads to overly large, relatively
uninformative CIs. For example, we implemented the ridge regression CI from [Thm. 3, 11] for our `FlightDelays`
experiment (an implementable CI is not provided for any other learning algorithm). This CI takes as input the maximum
absolute value of the target $y$ ($B_Y = 8.03$ after mean-centering) and the maximum $\ell_2$ norm of a feature vector $x$ (after
mean-centering, $B_X = 13.17$ with standardization or $B_X = 4200$ without). When standardizing as in Fig. 2, the
smallest width produced by [Thm. 3, 11] for any value of $n$ is 90.2; that is 86 times larger than the largest width of
our CLT intervals (equal to 1.04). When not standardizing as in App. K Fig. 3, our maximum width is 1.03, but the
minimum [Thm. 3, 11] width is $5 \times 10^{14}$. We will emphasize this important advantage of CLT intervals in the revision.

**(R1) Stability clarifications:** We will clarify in the revision that our stability assumptions
1. Do not require that $h_n$ be convex (in fact, many past stability results are for a 0-1 validation loss [12,15,16,19,4])
and do not require that $h_n$ be related to a loss function used to train a learning method
2. Cover $k$-nearest neighbor methods [12], decision tree methods [4], and ensemble methods [16] in addition to
non-convex SGD and strongly convex ERM
3. Hold even when training error is a poor proxy for test error due to overfitting (e.g., 1-nearest neighbor has training
error 0 but is still suitably stable [12])

We are not aware of other approaches that provide CLTs for such a broad class of learning algorithms and losses, but we
would appreciate any pointers to literature that we have missed.

**(R1) Experiments:** We appreciate the suggestions to improve our presentation and will introduce a new experiment
with synthetic data generated from a known model. We feel it is important to also maintain our existing real-data
experiments, as these best reflect how the competing CIs and tests perform in practice, under the eccentricities of
real data which are hard to capture with synthetic data. For example, it is common in real data to have one method
dominate for smaller sample sizes and the other dominate for larger sample sizes; this is precisely what we see in
the right column of Fig. 2. We will clarify that the aim of this assessment is not to establish power convergence or
to assess power in an absolute sense but rather to verify that for a diversity of settings encountered in the wild (e.g.,
random forest much better than logistic regression in Fig. 2 left and ridge regression barely better than neural network
in Fig. 2 right), our tests provide power as good as (and often better than) the most popular heuristics from the literature.
We will clarify that the reported sizes $= \frac{\text{\# of rejected } H_0 \text{ simulations}}{\text{\# } H_0 \text{ simulations}}$ and powers $= \frac{\text{\# of rejected } H_1 \text{ simulations}}{\text{\# } H_1 \text{ simulations}}$, where one of the
500 simulations is declared $H_0$ if the test error of $\mathcal{A}_2 \leq$ test error of $\mathcal{A}_1$ and $H_1$ otherwise. Notably, we only see size
estimates exceeding the level when the number of $H_0$ simulations is very small (when $\mathcal{A}_2$ improves upon $\mathcal{A}_1$ in so few
simulation replications that the Monte Carlo error in the size estimate is large).

**(R1) Known $R_n$:** In addition, we have rerun all `FlightDelays` regression experiments using an exact known $R_n$
(so that $R_n$ need not be estimated). In these new experiments, we take the population distribution to be the empirical
distribution over our entire `FlightDelays` dataset (so that $R_n$ is an expectation over 5.8M datapoints) and sample
training sets independently from this population. With this setup, we can exactly determine which of two algorithms
has better $k$-fold test error, and the results are comfortingly very similar to those reported in the submission.

**(R1) $V_n$:** We have only studied the conditional setting, but [5] recently proved an unconditional CLT under much more
restrictive assumptions than their conditional CLT and found consistent variance estimation to be more elusive.

**(R1) Terminology:** In the revision, we will clarify the formal definitions of "asymptotically exact" (coverage converging
to *exactly* $1 - \alpha$) and "asymptotically valid" (coverage asymptotically $\geq 1 - \alpha$), as the former is a stronger property.

**(R3) Non-asymptotic linearity:** In the revision, we will highlight that (3.2) in Thm. 2 already implies a "non-
asymptotic linearity" statement by providing an explicit non-asymptotic bound on the departure from linearity in terms
of the algorithm's loss stability: $\mathbb{E}[(\frac{\sqrt{n}}{\sigma_n}(\hat{R}_n - R_n) - \frac{1}{\sigma_n \sqrt{n}} \sum_{i=1}^{n} (\bar{h}_n(Z_i) - \mathbb{E}[\bar{h}_n(Z_i)]))^2] \leq \frac{3}{2\sigma_n^2} n(1 - \frac{1}{k}) \gamma_{loss}(h_n)$.

**(R3) AMSE:** We will clarify in the revision that we have no particular interest in unbiasedness; rather, our interest
in CV comes from its popularity: consumers and developers of ML methods are already using CV to estimate test
error, and we aim to turn those readily available estimates into valid inferences about test error without requiring any
new expensive computation (i.e., using only standard CV outputs). In addition, for estimating the mean of a univariate
normal, the best unbiased estimator is admissible and minimax optimal, so while bias can improve MSE for some
values of the true mean, no alternative estimator will have better MSE for all values of the unknown true mean.

**(R4) Our weaker assumptions:** In the revision, we will endeavor to improve intuition, highlighting that past results
exclude asymmetric (like SGD), inconsistent, and less stable learning algorithms and heavy-tailed data distributions
(see Apps. F & G for detailed examples of simple learning problems excluded by past work but covered by ours).

[Meta-Review · NeurIPS 2020]

The reviewers were all rather positive about the theoretical contribution, although one minority negative review (R1) gave a low score due an the experimental setup deemed unconvincing. Overall I recommend acceptance, possibly asking the authors to make some revisions to the experimental section to address some criticisms of R1.